# UEP: Unifying Estimation and Prediction for Non-stationary Multi-armed Bandits

## Abstract

Non-stationary multi-armed bandits present a fundamental challenge in sequential decision-making due to evolving reward distributions. Existing statistical estimation-based work often overlooked the learnable temporal patterns inherent in many real-world applications that encode valuable information for future trend prediction. To leverage such patterns, we propose a unified framework - UEP - to capture these dynamic patterns with a combination of both statistical estimation(estimation) and predictive model(predictor). According to estimation errors, UEP automatically determines optimal window sizes and the mixing weights in balancing predictor and estimator through an adaptively calculated weight, without requiring prior environmental knowledge. We prove regret bounds of $O(K^{(3d+2/2d+1)}T^{(d+1)/(2d+1)}(\log(KT))^{1/2}$. It improves upon existing $O(K^{1/3}T^{1-d/3})$ results when the environment changes fast at $d < 1$ under mild assumptions. With a series of experiments, we demonstrate both the efficacy of our algorithm and the broader applicability of our techniques to the complex, rapidly evolving time series.

## 1 Introduction

The multi-armed bandit (MAB) problem represents a fundamental challenge in sequential decision-making, where an agent must balance exploration of uncertain options with exploitation of currently promising alternatives. This framework has proven invaluable across diverse applications, e.g. clinical trial design, online advertising, recommendation systems, etc. (Slivkins et al., 2019). The most widely studied version of this model is the stochastic MAB, which assumes that the reward for each arm is drawn from an unknown but fixed distribution. Classical algorithms such as Upper Confidence Bound (UCB) (Auer et al., 2002) and Thompson Sampling (Thompson, 1933) have achieved the optimal $O(K \log T / \Delta_{\min})$ regret bounds under this stationary reward distribution condition.

Existing approaches in non-stationary MAB typically handle temporal changes through three main strategies. Some methods gradually forget historical information using sliding windows (Garivier & Moulines, 2011) or exponential discounting (Kocsis & Szepesvári, 2006) which cannot handle fast temporal changes. Others focus on detecting distribution shifts through statistical tests, such as CUSUM-based detection (Liu et al., 2018). More recent approaches combine multiple techniques but often require restrictive environmental assumptions—for instance, Komiyama et al. (2024) they assume that non-stationary environments will eventually converge to stationary states, particularly those with gradually changing rewards, limiting applicability to persistently changing environments without convergence guarantees. Despite these different strategies, existing methods generally share a common design philosophy: they normally use certain statistical models based on historical data to estimate the reward distributions, which might fast change in non-stationary environments.

However, we argued, most dynamic environments, albeit non-stationary, contain learnable temporal patterns—trends, periodicities, or systematic variations which are the foundation for the system's future prediction (Pourahmadi, 2001). Thus, rather than only use those statistic-based estimations, *can we also model and exploit those time-variation patterns for prediction*? To address this question, this paper introduces UEP, a unified framework that combines backward statistical estimation with forward predictive modeling to enable proactive adaptation in non-stationary environments. A probabilistic time series forecasting component based on the diffusion model is proposed to forecast reward distributions from historical observations, generating anticipated reward estimates for future

time steps. These predictions are then dynamically fused with traditional statistical estimations, according to prediction qualities. Furthermore, an adaptive weight parameter $\lambda_{i,t}$ is proposed to dynamically adjust the weight between estimation and prediction based on online prediction qualities. This design creates a unified approach between purely statistical strategies ($\lambda_{i,t} = 0$) and prediction-enhanced decision-making ($\lambda_{i,t} > 0$). The main contributions are:

1) **A unified framework**: We introduce UEP, which dynamically balances statistical estimation and predictive modeling through adaptive weight $\lambda_{i,t}$, enabling automatic adaptation between reactive and anticipatory strategies. 2) **Theoretical guarantee**: We prove $O(K^{(3d+2)/(2d+1)}T^{(d+1)/(2d+1)}(\log(KT)^{1/2})$ regret bounds that approach the theoretical lower bound $\Omega(K^{1/3}T^{1-d/3})$. 3) **Adaptive parameter selection**: We develop optimal window sizing and mixing weight selection requiring no prior environmental knowledge.

Comprehensive evaluations demonstrate the consistent performance across diverse environments, with superior results in structured settings and competitive performance in chaotic conditions.

## 2 RELATED WORK

**Non-stationary MAB:** Traditional approaches to non-stationary bandits rely on statistical estimation from historical observations, exhibiting *reactive* characteristics. The most established paradigm involves temporal weight schemes: SW-UCB (Garivier & Moulines, 2011) maintains sliding windows of fixed size achieving $O(\sqrt{KT^{1-d/2}})$ regret bounds, while D-UCB (Kocsis & Szepesvári, 2006) employs exponential discounting for computational efficiency. More sophisticated approaches detect distribution shifts through statistical tests—CUSUM-UCB (Liu et al., 2018) employs cumulative sum statistics while Page-Hinkley methods (Hartland et al., 2007) use sequential hypothesis testing. These methods demonstrate the core trade-off in backward estimation: larger windows provide stability but slow adaptation, while smaller windows enable rapid response but increase variance. Recent hybrid frameworks have combined multiple statistical techniques, with ADR-bandit (Komiyama et al., 2024) achieving optimal $O(\log T/\Delta_{min})$ regret in non-stationary settings with $d > 1$ while maintaining $O(\sqrt{MT})$ bounds with the assumption of the coordinated arm changes. While these methods have achieved significant success, they generally limit on using pre-defined models to estimate distributions. One recent approach, assumes the rewards follow the auto-regressive (AR) structure to mitigate this limitation Chen et al. (2023). However, this assumption is not valid in many real-world applications.

**Time Series Prediction Methods:** Temporal data naturally exhibits learnable patterns—trends, periodicities, and systematic variations—that enable predictive modeling (Wu et al., 2021). Unlike predefined statistical estimators that assume fixed distributional forms, time series forecasting methods offer greater flexibility by learning adaptive representations directly from sequential observations (Zhou et al., 2021). Modern approaches span diverse architectures including transformer-based models (Liu et al., 2022), convolutional networks with temporal convolutions (Wu et al., 2023), and hybrid architectures combining multiple mechanisms (Nie et al., 2023). For sequential decision-making contexts, the critical requirement extends beyond point estimation to full distributional forecasting, providing both expected values and uncertainty quantification essential for exploration-exploitation balance (Salinas et al., 2020; Rangapuram et al., 2018). Diffusion models have emerged as particularly promising for this task, offering natural distribution matching capabilities through iterative denoising processes (Tashiro et al., 2021; López Alcaraz & Strodthoff, 2022) and demonstrating superior performance in capturing complex temporal dependencies with principled uncertainty estimation. Recent work has begun exploring the integration of distributional forecasts into control and decision-making systems (Kostrikov et al., 2022; Janner et al., 2022), though systematic incorporation into multi-armed bandit algorithms remains largely unexplored.

## 3 PROBLEM FORMULATION

**Non-stationary multi-armed bandit problem:** Consider an agent facing $K$ arms over time horizon $T$, where each arm $i \in \{1, \ldots, K\}$ provides rewards $r_{i,t} \in [0, 1]$ drawn from time-varying distributions with mean $\mu_{i,t}$ and variance $\sigma_i^2$ at time $t \in \{1, \ldots, T\}$. At each time step, the agent selects exactly one arm $I_t$ and observes the corresponding reward $r_{I_t,t}$. For non-stationary environ-

ments, we assume the expected rewards $\mu_{i,t}$ is arm-independent and evolve over time according to the following characterization:

**Definition 1** (Environmental changes). *The rate of environmental change is characterized by parameter $d$ such that for any arm $i$ and time steps $s < t$:*

$$|\mu_{i,t} - \mu_{i,s}| \leq C \cdot \left(\frac{t-s}{t}\right)^d \quad \exists C > 0, \quad \forall i \in \{1, \ldots, K\}$$

*The deviation between observed reward $r_{i,t}$ and expected reward $\mu_{i,t}$ follows a Gaussian distribution $\mathcal{N}(0, \sigma_i^2)$, bounded by $\sigma_{\max} = \max_{i \in K} \sigma_i$.*

Here, $C$ is a constant and $d$ determines the change rate: larger $d$ indicates slower environmental changes. When $d > 1$, the environment changes smoothly with $\sum_{t=2}^{\infty} |\mu_{i,t} - \mu_{i,t-1}|$ convergent. When $0 < d \leq 1$, changes are rapid with divergent cumulative variation. When $d \approx 0$, differences between adjacent time steps can approach $C$, reflecting highly erratic changes.

**Rewards Estimation and Arm Selection:** The core challenge lies in estimating the time-varying expected rewards $\mu_{i,t}$ to guide arm selection by balancing exploration and exploitation.

**Definition 2** (Estimation and ARM selection). *Let $\mathcal{A}$ be an algorithm that, at each round $t \in [T]$, estimates the predicted reward expectation $\hat{\mu}_{i,t}$ for all arm $i \in [K]$ and chooses arm $I_t$ to pull based on the history $\{I_1, r_1, \ldots, I_{t-1}, r_{t-1}\}$ and corresponding window $W_{i,t}$:*

$$\hat{\mu}_{i,t}, I_t = F_{\mathcal{A}}(\mathcal{H}_{i,t-1}, W_{i,t})$$

In stationary or slowly changing environments, $W_{i,t}$ can be a fixed-size window, as exemplified by the Sliding-Window UCB (Garivier & Moulines, 2011), which relies solely on the most recent $\tau$ observations. In a non-stationary environment, $W_{i,t}$ can be dynamic to enable rapid adaptation. The ADR bandit (Komiyama et al., 2024) employs a data-dependent adaptive window, allowing it to contract upon detecting a change point. Due to the uncertainty of the predictive model and the limited observation data with dynamic windows, the decision $I_t$ made by the algorithm should jointly consider both $\hat{\mu}_{i,t}$ and its corresponding mathematical metrics, such as confidence bounds.

**Optimization objective:** The performance is measured by cumulative regret, defined as the total difference between optimal and achieved rewards (Lattimore & Szepesvári, 2020; Slivkins et al., 2019). The goal is to design an algorithm $\mathcal{A}$ to minimize the accumulated regret:

$$R(T) = \arg\min_{\mathcal{A}} \sum_{t=1}^{T} (\mu_t^* - \mu_{I_t,t}) \tag{1}$$

where $\mu_t^* = \max_{i \in K} \mu_{i,t}$ is the optimal expected reward at time $t$.

**Our motivation:** As discussed in Definition 1, environments might have totally different dynamic patterns. For a given environment, we generally do not have the knowledge about $d$.

Traditional statistical models (Krishnamurthy & Gopalan, 2021; Jia et al., 2023) are normally designed for certain types of environments, e.g. slow changing or slow changing with abrupt changes. They are not suitable for general environments with unknown $d$, especially the rapidly changing environments ($0 < d \leq 1$). In comparison, pure prediction models alone may demand significant data for modeling and could be influenced by some noise. Moreover, some predictors (e.g., AR2 (Chen et al., 2023)) assume a structured generative process and may perform poorly in highly stochastic or non-stationary settings where patterns are weak or inconsistent. To address this issue, we propose the usage of the two models with the following equation.

**Definition 3** (Unified estimator). *$\hat{\mu}_{i,t}$ consists of estimator model $f_e(\mathcal{H}_{i,t})$ and predictor model $f_p(\mathcal{H}_{i,t})$ with an adaptive weight $\lambda_{i,t}$:*

$$\hat{\mu}_{i,t} = (1 - \lambda_{i,t}) f_e(\mathcal{H}_{i,t-1}, W_{i,t}) + \lambda_{i,t} f_p(\mathcal{H}_{i,t-1}) \tag{2}$$

*where $\mathcal{H}_{i,t-1}$ is historical data for arm $i$ to time $t-1$ and $f_e$ can be any statistical model.*

To adapt to non-stationary environment, dynamic windows are employed within statistical models for forecasting, thereby mitigating errors from outdated data. In contrast, the predictor utilizes a fixed-size window, which maintains model stability and avoids overfitting, thus balancing adaptation and generalization. Thus, many classical methods as special cases: statistical estimation ($\lambda_{i,t} \equiv 0$) and pure prediction ($\lambda_{i,t} \equiv 1$).

# 4 THEORY

In this section, we provide theoretical support for our designed framework and prove its effectiveness in different environment settings with mild conditions.

As discussed previously, UEP employs the parametric predictor $\hat{\mu}_{i,t}^{\text{pred}}$ as the *Predictor* $f_p(\mathcal{H}_{i,t-1})$. The accuracy of this predictor primarily depends on the amount of data available for each arm, since more observations of arm $i$ improve prediction quality by facilitating learning from arm-specific data. Thus, we formalize the predictor's effectiveness with the following assumption.

**Assumption 1** (Predictor effectiveness). *There exists a constant $C_p > 0$ such that the prediction accuracy is bounded above by a function inversely proportional to the number of observations.*

$$\mathbb{E}[|\hat{\mu}_{i,t}^{pred} - \mu_{i,t}|^2] \leq C_p \cdot N_{i,t}^{-1} \tag{3}$$

If an arm is selected too few times, the predictor for that arm will be highly inaccurate. Therefore, we make an assumption about the number of times arm $i$ is selected within a window:

**Assumption 2** (Guarantee exploration). *There exists a constant $C_{ge} > 0$ such that, as the window size increases, the selection count for arm $i$ follows:*

$$N_{i,t} \geq C_{ge}\frac{W_{i,t}}{K}$$

After determining the calculation method for $\hat{\mu}_{i,t}^{\text{esti}}$ and the prediction error of $\hat{\mu}_{i,t}^{\text{pred}}$, to obtain $\hat{\mu}_{i,t}$, we still need to determine the adaptive parameters $W_{i,t}$ and $\lambda_{i,t}$ that minimize the $R(T)$.

**Lemma 1** (Optimal window size). *Based on Assumption 1 and Assumption 2, the optimal $W_{i,t}^*$ minimizing $R(T)$ increases monotonically with the growth of $d$ and $t$:*

$$W_{i,t}^* = C_w K^{-1/(2d+1)} t^{2d/(2d+1)} \quad \forall N_{i,t} > 0 \tag{4}$$

*The constant $C_w$ serves as a balancing factor that encapsulates environmental dynamics and prediction accuracy, with its detailed derivation provided in Appendix A.5.*

*Proof.* (Details are provided in Appendix A.5) At moment $t$, to get the optimal $W_{i,t}$, we minimize the $R(T)$ and treat $MSE(\hat{\mu}_{i,t}, \mu_{i,t})$ as function $L(W_{i,t}, \lambda_{i,t})$. Since the true mean $\mu_{i,t}$ is unknown, direct minimization is infeasible. Instead, we construct a surrogate function $L_{\text{sub}}(W_{i,t}, \lambda_{i,t})$ that upper-bounds the true MSE. This surrogate consists of two parts: the first part bounds the MSE of the estimator, $MSE(\hat{\mu}_{i,t}^{\text{esti}}, \mu_{i,t})$, using the environmental smoothness condition (Definition 1) and the exploration guarantee (Assumption 2), yielding an upper bound of $\frac{\sigma_i^2 K}{C_{ge} \cdot W_{i,t}} + \frac{C^2 K^2 W_{i,t}^{2d}}{C_{ge}^2 (d+1)^2 t^{2d}}$. The second part bounds the MSE of the predictor, $MSE(\hat{\mu}_{i,t}^{\text{pred}}, \mu_{i,t})$, based on its assumed predictor effectiveness (Assumption 1), resulting in a bound of $\frac{C_p K}{C_{gx} \cdot W_{i,t}}$. Given the convexity of $L_{\text{sub}}$ with respect to $W_{i,t}$, we solve $\frac{\partial L_{sub}}{\partial W_{i,t}} = 0$ to derive the optimal window size $W_{i,t}^*$. □

According to Eq. 4, as time progresses, the impact of drift diminishes, allowing larger windows for more stable estimates. Conversely, smaller $d$ values (faster changes) require smaller windows to maintain accuracy.

Note that $W_{i,t}^*$ is derived by minimizing a surrogate upper bound of the $MSE(\hat{\mu}_{i,t}, \mu_{i,t})$, not the exact $MSE(\hat{\mu}_{i,t}, \mu_{i,t})$. Therefore, $W_{i,t}^*$ is sufficient to ensure that the $MSE(\hat{\mu}_{i,t}, \mu_{i,t})$ is bounded optimally, but it is not necessary for achieving the minimum possible $MSE(\hat{\mu}_{i,t}, \mu_{i,t})$. Other window sizes may also lead to the same or similar $MSE(\hat{\mu}_{i,t}, \mu_{i,t})$ values. However, under our smoothness and bounded-error assumptions, $W_{i,t}^*$ is order-optimal.

**Lemma 2** (Optimal weight). *Based on the definition of $\hat{\mu}_{i,t}$(Eq. 2), $\hat{\mu}^{esti}$(Eq. 6) and prediction model capability(Eq. 3):*

$$\lambda_{i,t}^* = \frac{A(W_{i,t}) - C(W_{i,t})}{A(W_{i,t}) + B(W_{i,t}) - 2C(W_{i,t})} \tag{5}$$

where: $A(W_{i,t}) = MSE(\hat{\mu}_{i,t}^{esti}, \mu_{i,t})$ *is the mean squared error of the estimator,* $B(W_{i,t}) = MSE(\hat{\mu}_{i,t}^{pred}, \mu_{i,t})$ *is the mean squared error of the predictor and* $C(W_{i,t}) = E[(\hat{\mu}_{i,t}^{esti} - \mu_{i,t})(\hat{\mu}_{i,t}^{pred} - \mu_{i,t}))]$ *is the cross-error term.*

*Proof.* (Details are provided in Appendix A.6) Similar to the proof of Lemma 1, we also minimize the loss function $L(W_{i,t}, \lambda_{i,t})$ under fixed parameters $t$ and $W_{i,t}$. As $L(W_{i,t}, \lambda_{i,t})$ is convex in $\lambda_{i,t}$, let $\frac{\partial L}{\partial \lambda_{i,t}} = 0$ to get $\lambda_{i,t}^*$. $\qquad\square$

According to Eq. 5, in stable environments where historical data remains highly informative, the estimation component typically achieves lower $MSE$ than prediction due to reduced environmental drift, leading to $\lambda_{i,t}^* \to 0$ and recovery of classical UCB-style behavior.

It is important to note that $\lambda_{i,t}^*$ is both necessary and sufficient to achieve the minimum $MSE(\hat{\mu}_{i,t}, \mu_{i,t})$ for a given $W_{i,t}$. When combined with the (sufficient) choice of $W_{i,t}^*$, the pair $(W_{i,t}^*, \lambda_{i,t}^*)$ is sufficient for achieving near-optimal MSE performance.

With the parameters above, we can obtain $\hat{\mu}_{i,t}$. To select the optimal arm, we then define a score $SCORE_{i,t} = \hat{\mu}_{i,t} + CB_{i,t}$ (a detailed discussion is provided in Section 5.2.3) and need to calculate the confidence bound $CB_{i,t}$.

## 4.1 QUANTIFYING THE PREDICTION UNCERTAINTY

Due to potential errors introduced by the prediction model and the selection of window size, we introduce the confidence bound $CB_{i,t}$ to quantify the uncertainty in the estimation. From the mixed estimator definition of $\hat{\mu}_{i,t}$(Eq. 2), we assert with probabilistic guarantee $(1 - \frac{\delta}{KT})$ that the estimation error $|\hat{\mu}_{i,t} - \mu_{i,t}|$ remains bounded by $CB_{i,t}$. This imposes a constraint on the prediction accuracy:

$$|\hat{\mu}_{i,t} - \mu_{i,t}| \leq CB_{i,t}(\forall i \in [K], t \in [T])$$

$CB_{i,t}$ can be obtained using mathematical formulas such as Hoeffding's inequality.

**Lemma 3** (Confidence bound). *Based on the definition of* $\hat{\mu}_{i,t}$*(Eq. 2),* $\hat{\mu}^{esti}$*(Eq. 6, defined later in Section 5.2.1) and predictor effectiveness(Assumption 1), the confidence bound is:*

$$CB_{i,t} = \sqrt{\frac{2\sigma_i^2 \log(4KT/\delta)}{N_{i,t}}} + ((1 - \lambda_{i,t})C_{e1} + \lambda_{i,t}C_{e2})K^{\frac{d+1}{2d+1}} t^{-\frac{d}{2d+1}}$$

*where* $C_{e1}^2 = \frac{\sigma_i^2}{C_{ge}C_w} + \frac{C^2 C_w^{2d}}{(d+1)^2 C_{ge}^2}$ *and* $C_{e2}^2 = \frac{C_p}{C_{ge}C_w}$

*Proof.* (Details are provided in Appendix A.7) Decompose $\hat{\mu}_{i,t} - \mu_{i,t}$ into two parts:

$$\hat{\mu}_{i,t} - \mu_{i,t} = (1 - \lambda_{i,t})(\hat{\mu}_{i,t}^{esti} - \mu_{i,t}) + \lambda_{i,t}(\hat{\mu}_{i,t}^{pred} - \mu_{i,t})$$

$$= [(1 - \lambda_{i,t})(\hat{\mu}_{i,t}^{esti} - \mathbb{E}[\hat{\mu}_{i,t}^{esti}]) + \lambda_{i,t}(\hat{\mu}_{i,t}^{pred} - \mathbb{E}[\hat{\mu}_{i,t}^{pred}])]$$

$$+ [(1 - \lambda_{i,t})(\mathbb{E}[\hat{\mu}_{i,t}^{esti}] - \mu_{i,t}) + \lambda_{i,t}(\mathbb{E}[\hat{\mu}_{i,t}^{pred}] - \mu_{i,t})]$$

First based on predictor effectiveness(Assumption 1) and using Hoeffding inequality calculate $[(1 - \lambda_{i,t})(\hat{\mu}_{i,t}^{esti} - \mathbb{E}[\hat{\mu}_{i,t}^{esti}]) + \lambda_{i,t}(\hat{\mu}_{i,t}^{esti} - \mathbb{E}[\hat{\mu}_{i,t}^{esti}])]$. Then use determinism to determine the scope of $\mathbb{E}[\hat{\mu}_{i,t}^{esti}] - \mu_{i,t}$ and $\mathbb{E}[\hat{\mu}_{i,t}^{pred}] - \mu_{i,t}$. $\qquad\square$

## 4.2 REGRET BOUND

We use the expected cumulative regret $\mathbb{E}[R(T)]$ in Eq. 1 to evaluate the performance of the algorithm.

**Theorem 1.** *In the non-stationary MAB problem, use the UEP algorithm, then the expected cumulative regret satisfies:*

$$\mathbb{E}[R(T)] = O\left(K^{(3d+2/2d+1)}T^{(d+1)/(2d+1)}(\log(KT))^{1/2}\right)$$

*Proof.* (A detailed proof is provided in Appendix A.8.) Using the optimal parameters $W_{i,t}^*$ selected as in Lemma 1 and $\lambda_{i,t}^*$ are selected as shown in Lemma 2, we compute the one-step regret $\mathbb{E}[r_t]$ by analyzing both the failure and effectiveness scenarios of the confidence interval, resulting in $\mathbb{E}[r_t] \leq \sum_{i=1}^{K} \mathbb{P}(|\mu_{i_t^*,t} - \mu_{i,t}| \leq 2\mathrm{CB}_{i^*,t} + 2\mathrm{CB}_{i,t}) \cdot O(\max_{i \in \{1,\dots,K\}}(\mathrm{CB}_{i,t})) + O(\frac{1}{T})$. By subsequently summing this one-step regret over the time horizon $T$, we derive the total expected cumulative regret $\mathbb{E}[R(T)]$. $\qquad\square$

When $d \geq 1$, the exponent $(d+1)/(2d+1)$ approaches the theoretical lower bound $\Omega(K^{1/3}T^{1-d/3})$(In Appendix A.9). When $d < 1$, the exponent $(d+1)/(2d+1) < 1 - (d/3)$ is superior to classical methods (regret=$O(\sqrt{K}T^{1-(d/3)})$) in rapidly changing environments.

# 5 OUR METHOD

In this section, we introduce the proposed UEP framework, including the estimator model, the predictor model, as well as the mixed weight mechanism.

## 5.1 GENERAL ALGORITHM

UEP implements a theoretically-grounded mixing framework that combines statistical estimation and predictive modeling through optimal weights derived in Lemma 2. The algorithm dynamically computes adaptive window sizes $W_{i,t}^*$ for each arm using the theoretical formula from Lemma 1, ensuring optimal bias-variance trade-offs as environmental conditions evolve. Statistical estimates $\hat{\mu}_{i,t}^{\mathrm{esti}}$ are computed from recent observations within these adaptive windows, while predictive estimates $\hat{\mu}_{i,t}^{\mathrm{pred}}$ are generated through arm-specific diffusion models that learn temporal patterns from historical reward sequences. Final arm selection employs mixed estimates combined with rigorous confidence bounds from Lemma 3, ensuring theoretical regret guarantees while exploiting learnable temporal structures.

---

**Algorithm 1** UEP

1: **Initialize:** $K$ arms, diffusion models $\{f_p^{(i)}\}$, $\hat{d} = 1$

2: **for** $t = 1$ to $T$ **do**
3:     **if** $t \bmod \lceil \sqrt{t} \rceil = 0$ **then**
4:         Update $\hat{d}$ from historical data
5:     **end if**
6:     $W_{i,t} \leftarrow$ optimal window (Lemma 1)
7:     **for** each arm $i$ **do**
8:         Compute $\hat{\mu}_i^{\mathrm{esti}}$ from recent $W_{i,t}$ samples
9:         Sample $\hat{\mu}_i^{\mathrm{pred}} \sim f_p^{(i)}(\mathcal{H}_{i,t-1})$
10:       Estimate MSEs and covariance bound
11:       $\lambda_i \leftarrow$ optimal weight (Lemma 2)
12:       $\hat{\mu}_i \leftarrow (1-\lambda_i)\hat{\mu}_i^{\mathrm{esti}} + \lambda_i\hat{\mu}_i^{\mathrm{pred}}$
13:       $\mathrm{SCORE}_i \leftarrow \hat{\mu}_i + \mathrm{CB}_i$ (Lemma 3)
14:     **end for**
15:     Select $I_t = \arg\max_i \mathrm{SCORE}_i$, observe $r_t$
16:     Update $f_p^{(I_t)}$ with new observation
17: **end for**

---

## 5.2 KEY MODULES

### 5.2.1 WINDOW-BASED ESTIMATOR

$f_e(\mathcal{H}_{i,t})$ performs statistical analysis on historical data to estimate $\mu_{i,t}$. Previous works have provided a variety of $f_e(\mathcal{H}_{i,t})$. Without loss of generality, UEP adopts a UCB-like definition $\hat{\mu}_{i,t}^{\mathrm{esti}}$ as $f_e(\mathcal{H}_{i,t})$:

$$\hat{\mu}_{i,t}^{\mathrm{esti}} = \frac{1}{N_{i,t}} \sum_{s=\max(1,t-W_{i,t}),I_s=i}^{t-1} r_{I_s,s} \tag{6}$$

where $\hat{\mu}_{i,t}^{\mathrm{esti}}$ represents the statistical reward estimate for arm $i$ at time $t$, computed as the average of historical rewards within an adaptive window of size $W_{i,t}$. Here, our definition differs from SW-UCB in which $W_{i,t}$ is fixed. In our solution, $W_{i,t}$ is the adaptive window size and will be calculated at each time $t$ and each arm $i$ defined in the theory section.

Since arm $i$ is not selected at every time step, $N_{i,t}$ denotes the number of times arm $i$ was selected within the current window, $I_s$ represents the arm selected at time step $s$, and $r_{I_s,s}$ is the correspond-

ing observed reward. This approach captures recent performance trends while adapting the window size to environmental changes. Our definition is related to traditional UCB algorithms:

- UCB: $\lambda_{i,t} \equiv 0$, $W_{i,t} = t - 1$ (uses all data)
- SW-UCB: $\lambda_{i,t} \equiv 0$, fixed $W_{i,t}$ (uses fixed sliding window)

In non-stationary MAB problem, when the window is too large, distant history becomes noise; when the window is too small, the lack of history may affect the estimation results. Compared to all-historical windows and fixed-size windows, the UEP algorithm dynamically adjusts the window size to adaptively capture environmental changes.

### 5.2.2 DIFFUSION-BASED PREDICTOR

The predictor $f_p(\mathcal{H}_{i,t-1})$ aims to learn the conditional reward distribution for each arm by exploiting temporal patterns in historical observations. We implement the predictor using Denoising Diffusion Probabilistic Models (DDPM) (Ho et al., 2020), which learn to model complex distributions through an iterative denoising process.

The forward diffusion process corrupts reward observations through a variance schedule:
$$q(y_t|y_0) = \mathcal{N}(y_t; \sqrt{\bar{\alpha}_t}y_0, (1 - \bar{\alpha}_t)\mathbf{I})$$
where $y_0$ represents the true reward and $\bar{\alpha}_t = \prod_{s=1}^{t} \alpha_s$ with $\alpha_t = 1 - \beta_t$. We employ a cosine noise schedule $\beta_t$ for stable training across $T = 20$ timesteps.

The conditional denoising network learns to reverse this process:
$$\epsilon_\theta(y_t, t, \mathcal{H}_{i,t-1}, i) = \text{NoisePredictor}(\text{concat}[y_t, \text{TimeEmb}(t), \text{ContextEmb}(\mathcal{H}_{i,t-1}), \text{ArmEmb}(i)])$$

The predictor generates both point estimates and uncertainty quantification:
$$\hat{\mu}_{i,t}^{\text{pred}} = \frac{1}{\sqrt{\alpha_t}}\left(y_t - \frac{\beta_t}{\sqrt{1 - \bar{\alpha}_t}}\epsilon_\theta\right)$$
$$\hat{\sigma}_{i,t}^{\text{pred}} = \text{VarianceEstimator}(\mathcal{H}_{i,t-1}, t, i)$$

However, the sparse observation problem inherent in multi-armed bandits significantly challenges the predictor's effectiveness. Since arm $i$ is only selected when $I_s = i$, the number of observations $N_{i,t}$ for each arm is typically much smaller than the total time horizon ($N_{i,t} \ll t$). This sparsity directly impacts the predictor's ability to learn meaningful temporal patterns, as reflected in Assumption 1. To address this challenge, our implementation employs arm-independent modeling where each arm maintains separate historical buffers and step counters. The predictor uses a conservative window sizing strategy to ensure stable learning:
$$\text{window\_size}_{\text{pred}}[i] = \min(\text{default\_window}, |\text{hist\_buffers}[i]|)$$

### 5.2.3 ARM SELECTION

The optimal arm is selected through a two-component scoring system where the mixed prediction $\hat{\mu}_{i,t}$ is combined additively with its confidence bound $CB_{i,t}$:
$$SCORE_{i,t} = \hat{\mu}_{i,t} + CB_{i,t}$$
where $CB_{i,t}$ is the confidence bound that accounts for the uncertainty in our mixed prediction $\hat{\mu}_{i,t}$. The confidence bound incorporates both the uncertainty from the estimator and the predictor error, ensuring theoretical guarantees for our mixed approach. At each time step $t$, we select the arm $I_t$ with the highest score:
$$SCORE_{I_t,t} = \text{MAX}(SCORE_{i,t}, i \in \{1, \ldots, K\})$$
Detailed construction of $CB_{i,t}$ and theoretical analysis are provided in section 4.1.

## 6 EXPERIMENTS

### 6.1 EXPERIMENTAL SETUP

**Compared environments.** Figure 1 illustrates the four non-stationary experimental environments with varying degrees of non-stationarity. The **Smooth** environment ($d = 1.2$) features

learnable periodic patterns ideal for testing prediction capabilities, while **High Frequency** ($d = 0.6$) and **Abrupt** ($d = 0.8$) environments challenge rapid adaptation and change detection respectively. The **Competitive Balanced** environment ($d = 0.7$) cyclically alternates which arm is optimal over time. These environments, along with the stationary baseline environment, target distinct validation objectives. Detailed mathematical definitions of all experimental environments are provided in Appendix B.5.

**Baselines** To comprehensively evaluate UEP's performance, we selected 22 representative algorithms spanning four major paradigms in multi-armed bandit research: *classical confidence bound methods* (4 algorithms) including UCB1 (Auer

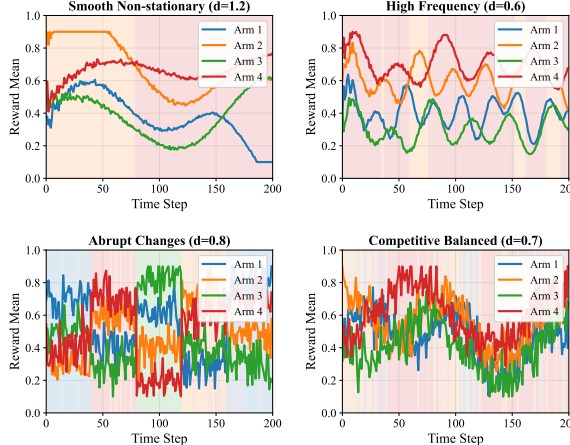

Figure 1: Reward patterns of four non-stationary environments with different smoothness parameters $d$.

et al., 2002), MOSS (Audibert & Bubeck, 2009), KL-UCB (Garivier & Cappé, 2011), and Variance-Aware UCB (Audibert et al., 2009); *Bayesian and sampling methods* (5 algorithms) including Beta-SWTS (Mellor & Shapiro, 2013); *non-stationary adaptation methods* (6 algorithms) including Sliding-Window UCB (Garivier & Moulines, 2011), Discounted UCB (Kocsis & Szepesvári, 2006); *change detection and exploration methods* (7 algorithms) including Change-Point-Detection (Liu et al., 2018). All 22 baselines provided in the appendix.

**Configuration** All experiments use $K = 4$ arms over $T = 1000$ time steps with each configuration averaged over 25 runs.

### 6.2 EXPERIMENT RESULTS

Table 1 presents the top-performing representatives from each category with 8 representative baselines, selected from four methodological paradigms. Due to page limits, complete comparison results against 22 strong baselines are provided in Table 4 in the appendix.

| Algorithm | Static | Smooth | Abrupt | High-Freq | Balanced | AVG Rank |
|---|---|---|---|---|---|---|
| **UEP** | **16.9±0.6** | **39.1±2.1** | **35.7±7.4** | **28.8±0.6** | 76.4±7.1 | 1.2 |
| Variance-Aware-UCB | 22.3±0.2 | 40.6±0.6 | 52.9±4.1 | 30.2±0.8 | 76.9±3.2 | 2.8 |
| MOSS | 24.2±0.4 | 41.0±0.5 | 57.7±4.4 | 32.5±0.4 | **71.3±3.9** | 3.2 |
| Change-Point-Detection | 17.8±0.1 | 62.8±2.4 | 49.1±17.6 | 47.6±1.9 | 90.9±9.6 | 4.6 |
| KL-UCB | 36.3±0.3 | 41.6±0.5 | 59.8±2.6 | 34.1±0.4 | 79.2±2.8 | 4.8 |
| UCB1 | 65.7±0.4 | 59.4±0.3 | 75.1±3.0 | 57.0±0.3 | 87.8±2.0 | 6.4 |
| Sliding-Window-UCB | 33.4±0.3 | 63.2±0.3 | 95.2±9.8 | 42.7±0.4 | 92.4±3.0 | 6.6 |
| Beta-SWTS | 18.5±1.0 | 81.0±3.2 | 160.8±18.0 | 39.0±1.6 | 97.1±3.8 | 6.6 |
| Discounted-UCB | 130.6±0.5 | 121.8±0.2 | 134.8±1.4 | 117.1±0.3 | 118.2±1.0 | 8.8 |

Table 1: Baseline Comparison: Cumulative Regret (Mean±Std)

UEP's consistent high-ranking performance across all environments validates the effectiveness of adaptive paradigm fusion, achieving 32.5% improvement over Sliding-Window-UCB in high-frequency scenarios while maintaining competitive performance in stable settings. This success stems from addressing fundamental limitations of existing paradigms. Sliding-Window-UCB employs a fixed sliding window for statistical estimation, providing robust theoretical guarantees, but its reactive-only adaptation cannot adjust window sizes to varying rates of environmental change. Change-Point-Detection excels in static environments (17.8) where its binary detection mechanism avoids false alarms, but exhibits poor performance in environments with gradual or continuous changes (62.8-90.9) where its assumption of discrete breakpoints fails and either misses smooth transitions or generates excessive false detections. Our UEP automatically adapts between statistical estimation and predictive enhancement based on prediction quality, eliminating the traditional

paradigm selection dilemma where specialized methods excel only under specific environments by discovering optimal mixing through online estimation.

## 6.3 ANALYSIS EXPERIMENTS

### 6.3.1 ADAPTIVE WEIGHT ANALYSIS

| Steps | d = 0.7 | | | d = 1.0 | | | $d \to \infty$ | | |
|---|---|---|---|---|---|---|---|---|---|
| | Pred Err (Var) | Est Err (Var) | $\lambda$ (Var) | Pred Err (Var) | Est Err (Var) | $\lambda$ (Var) | Pred Err (Var) | Est Err (Var) | $\lambda$ (Var) |
| 161-180 | 0.0766 (0.0030) | 0.1066 (0.0072) | 0.7903 (0.0386) | 0.1144 (0.0051) | 0.1112 (0.0048) | 0.6975 (0.0601) | 0.0813 (0.0034) | 0.0150 (0.0001) | 0.4528 (0.0999) |
| 181-200 | 0.0674 (0.0018) | 0.0498 (0.0011) | 0.7665 (0.0327) | 0.0970 (0.0068) | 0.0426 (0.0010) | 0.6810 (0.0768) | 0.0826 (0.0034) | 0.0178 (0.0002) | 0.4128 (0.1072) |
| **Average** | **0.0720 (0.0024)** | **0.0782 (0.0042)** | **0.7784 (0.0357)** | **0.1057 (0.0060)** | **0.0769 (0.0029)** | **0.6893 (0.0685)** | **0.0820 (0.0034)** | **0.0164 (0.0002)** | **0.4328 (0.1036)** |

Table 2: Adaptive weight $\lambda_{i,t}$ evolution across different environmental conditions.

Taking the high frequency environment as an example, we change the speed of environmental changes by influencing $d$. After reaching convergence, from Table 2 we can observe that when the environment changes rapidly, i.e., when the value of $d$ is small, the predictor's prediction error is significantly smaller than the window's estimation error, and decisions rely more on prediction, which conforms to Lemma 2. We are also pleasantly surprised to find that when the environment is equivalent to a steady state, the predictor still maintains good accuracy, but obviously the estimation error is smaller in this case, and decisions rely more on estimation.

### 6.3.2 COMPONENT CONTRIBUTION

Systematic ablation study reveals how individual components interact within the unified framework.

| Component | Setting | Static | High-Freq | Balanced |
|---|---|---|---|---|
| **Complete UEP** | Full framework | 20.2 | **30.3** | **75.5** |
| -AdaptiveWindow | Fixed window size = 20 | **18.1** (+10.4%) | 54.1 (-78.5%) | 104.2 (-38.0%) |
| -DiffusionModel | $\lambda_{i,t} = 0$ | 29.0 (-43.6%) | 63.1 (-108.2%) | 77.7 (-3.0%) |
| -TheoryGuided$\lambda$ | $\lambda_{i,t} = 0.5$ | 20.3 (-0.5%) | 61.5 (-103.0%) | 76.0 (-0.7%) |

Table 3: Component ablation results

The ablation results validate the theoretical framework's component hierarchy and demonstrate how each element contributes to automatic adaptation. Removing adaptive windowing causes severe degradation in dynamic environments (+78.5% in high-frequency, +38.0% in balanced) while showing minor benefits in static settings (-10.4%), confirming Lemma 1's premise that optimal window sizing is essential for proper bias-variance trade-offs. Disabling the diffusion predictor consistently degrades performance (+43.6% to +108.2%), while removing theory-guided adaptive weighting shows minimal impact in stable environments but substantial degradation in dynamic settings (+103.0%), demonstrating that Lemma 2's adaptive mixing mechanism is crucial for automatically balancing estimation and prediction without prior environmental knowledge.

## 7 CONCLUSION

This paper introduces UEP, a unified framework that integrates predictive modeling with statistical estimation for non-stationary multi-armed bandits through adaptive weight $\lambda_{i,t}$. Our theoretical analysis establishes improved regret bounds of $O(K^{(3d+2)/(2d+1)}T^{(d+1)/(2d+1)}(\log(KT))^{1/2})$ for rapidly changing environments while eliminating environment-specific algorithm selection, thereby positioning predictive modeling as a valuable complement to classical bandit methods.

## 8 ETHICS STATEMENT

This work presents a theoretical and algorithmic contribution to non-stationary multi-armed bandits without involving human subjects or sensitive data collection. We identify the following ethical considerations:

**Algorithmic Applications:** Our proposed UEP framework is designed for sequential decision-making problems and may be applied in domains such as online advertising, recommendation systems, and clinical trial design. While these applications can provide societal benefits through improved decision-making efficiency, we acknowledge potential concerns regarding fairness and transparency in automated decisions.

**Fairness and Bias:** The UEP algorithm optimizes cumulative regret without explicitly modeling fairness constraints. In applications where arms represent different demographic groups or sensitive categories, practitioners should carefully evaluate potential discriminatory outcomes and consider incorporating fairness-aware modifications.

**Experimental Ethics:** All experiments were conducted using synthetic environments without real user data or human subjects. The experimental design ensures reproducible and controlled evaluation while avoiding potential privacy or consent issues.

**Responsible Deployment:** We encourage researchers and practitioners to consider the broader societal implications when deploying our method in real-world applications, particularly in high-stakes domains such as healthcare or finance where algorithmic decisions may significantly impact individuals.

## 9 REPRODUCIBILITY STATEMENT

To ensure full reproducibility of our results, we have made the following provisions:

**Theoretical Results:** All theoretical proofs are provided in detail in Appendices A.5 through A.8, including complete derivations for optimal window sizing (Lemma 1), adaptive weight selection (Lemma 2), confidence bounds (Lemma 3), and regret analysis (Theorem 1). Mathematical assumptions are clearly stated in Assumptions 1 and 2.

**Experimental Setup:** Section 5.1 provides comprehensive details of all experimental environments, including complete mathematical definitions in Appendix B.5. All experiments use standardized parameters: $K = 4$ arms, $T = 1000$ time steps, averaged over 25 runs with statistical significance reporting.

**Algorithm Implementation:** The UEP algorithm is fully specified in Algorithm 1 with detailed implementation of core components described in Section 4.2. The diffusion-based predictor architecture and training procedures are comprehensively documented in Appendix B, including network architectures, hyperparameters, and training protocols.

**Baseline Comparisons:** We evaluated against 22 established algorithms spanning four major paradigms, with complete results provided in Table 4 in the appendix. All baseline implementations follow standard configurations from their respective publications.

**Code Availability:** Complete source code for the UEP algorithm, all baseline implementations, experimental environments, and evaluation scripts will be made publicly available upon publication to ensure full reproducibility of our results.

## 10 LARGE LANGUAGE MODEL USAGE STATEMENT

We used large language models to assist with language improvement in this research work:

**Language Polishing:** Large language models (GPT-4 and Claude-3.5-Sonnet) were used solely to improve the grammar, sentence structure, and overall readability of the manuscript. This included refining word choices, correcting grammatical errors, and enhancing the clarity of expression throughout the paper.

**Scope Limitations:** LLMs were not used for any other aspects of this research. Specifically, they were not involved in:

- Research ideation or conceptual development
- Mathematical derivations or theoretical analysis
- Algorithm design or implementation
- Experimental design or data analysis
- Literature review or citation selection
- Scientific conclusions or interpretations

All core scientific contributions, including the UEP framework design, theoretical analysis, experimental evaluation, and research conclusions, were conceived, developed, and executed entirely by the human authors. The authors take full responsibility for all technical content and scientific claims in this work.

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

## A  DERIVIATION

### A.1  LOSS FUNCTION AND MSE OF MIXED ESTIMATOR

**Lemma 4.** *Based on the definition of mixed estimator $\hat{\mu}_{i,t}$ in Definition 3:*

$$L(W_{i,t}, \lambda_{i,t}) = MSE(\hat{\mu}_{i,t}, \mu_{i,t})$$
$$= (1 - \lambda_{i,t})^2 MSE(\hat{\mu}_{i,t}^{esti}, \mu_{i,t}) + \lambda_{i,t}^2 MSE(\hat{\mu}_{i,t}^{pred}, \mu_{i,t}) + 2\lambda_{i,t}(1 - \lambda_{i,t})\mathbb{E}[\epsilon_e \epsilon_p]$$

*where $\epsilon_e = \hat{\mu}_{i,t}^{esti} - \mu_{i,t}$ and $\epsilon_p = \hat{\mu}_{i,t}^{pred} - \mu_{i,t}$*

*Proof.* Based on definition of mixed estimator $\hat{\mu}_{i,t}$ in Definition 3, we have:

$$\hat{\mu}_{i,t} - \mu_{i,t} = (1 - \lambda_{i,t})\epsilon_e + \lambda_{i,t}\epsilon_p$$

Use $\text{MSE}(\hat{\mu}_{i,t})$ as the loss function:

$$
\begin{aligned}
L(W, \lambda) &= \text{MSE}(\hat{\mu}_{i,t}, \mu_{i,t}) \\
&= \mathbb{E}[((1 - \lambda_{i,t})\epsilon_e + \lambda_{i,t}\epsilon_p)^2] \\
&= \mathbb{E}[(1 - \lambda_{i,t})^2 \epsilon_e^2 + \lambda_{i,t}^2 \epsilon_p^2 + 2\lambda_{i,t}(1 - \lambda_{i,t})\epsilon_e \epsilon_p] \\
&= (1 - \lambda_{i,t})^2 \mathbb{E}[\epsilon_e^2] + \lambda_{i,t}^2 \mathbb{E}[\epsilon_p^2] + 2\lambda_{i,t}(1 - \lambda_{i,t})\mathbb{E}[\epsilon_e \epsilon_p] \\
&= (1 - \lambda_{i,t})^2 \text{MSE}(\hat{\mu}_{i,t}^{esti}, \mu_{i,t}) + \lambda_{i,t}^2 \text{MSE}(\hat{\mu}_{i,t}^{pred}, \mu_{i,t}) \\
&\quad + 2\lambda_{i,t}(1 - \lambda_{i,t})\mathbb{E}[(\hat{\mu}_{i,t}^{esti} - \mu_{i,t})(\hat{\mu}_{i,t}^{pred} - \mu_{i,t})]
\end{aligned}
$$

where $\mathbb{E}[\epsilon_e^2] = \text{MSE}(\hat{\mu}_{i,t}^{esti})$, $\mathbb{E}[\epsilon_p^2] = \text{MSE}(\hat{\mu}_{i,t}^{pred})$ . $\qquad\square$

### A.2  ANALYSIS OF UPPER BOUNDS FOR COMBINED ERROR

**Lemma 5.** *Based on Appendix A.1 and its proof, the $\mathbb{E}[\epsilon_e \epsilon_p]$ satisfies:*

$$|\mathbb{E}[\epsilon_e \epsilon_p]| \leq \frac{MSE(\hat{\mu}_{i,t}^{esti}, \mu_{i,t}) + MSE(\hat{\mu}_{i,t}^{pred}, \mu_{i,t})}{2}$$

*Proof.* By the exactation form of Cauchy-Schwarz inequality, we have:

$$
\begin{aligned}
|\mathbb{E}[\epsilon_e \epsilon_p]| &\leq \sqrt{\mathbb{E}(\epsilon_e^2)}\sqrt{\mathbb{E}(\epsilon_p^2)} \\
&= \sqrt{\text{MSE}(\hat{\mu}_{i,t}^{esti}, \mu_{i,t})}\sqrt{\text{MSE}(\hat{\mu}_{i,t}^{pred}, \mu_{i,t})} \\
&\leq \frac{\text{MSE}(\hat{\mu}_{i,t}^{esti}, \mu_{i,t}) + \text{MSE}(\hat{\mu}_{i,t}^{pred}, \mu_{i,t})}{2}
\end{aligned}
$$

$\qquad\square$

### A.3  MSE OF ESTIMATION MODEL

**Lemma 6.** *Based on the definition of $\hat{\mu}_{i,t}^{esti}$(Definition 3) and the bounded errors and Assumption 2, let $\mathcal{W}_{i,t} = \{s : \tau_{i,t} \leq s < t, I_s = i\}$ is the set of time points within the window where arm $i$ is selected, where $\tau_{i,t} = \max(1, t - W_{i,t})$ is the window start time. We have:*

$$MSE(\hat{\mu}_{i,t}^{esti}, \mu_{i,t}) \leq \frac{\sigma_i^2 K}{C_{ge} \cdot W_{i,t}} + \frac{C^2 K^2 W_{i,t}^{2d}}{C_{ge}^2 (d+1)^2 t^{2d}}$$

*Proof.* Based on the definition of $\hat{\mu}_{i,t}^{esti}$(Definition 3) and the bounded errors, we have:

$$\mathbb{E}[\hat{\mu}_{i,t}^{\text{esti}}|N_{i,t},\mathcal{W}_{i,t}] = \frac{1}{N_{i,t}} \sum_{s\in\mathcal{W}_{i,t}} \mu_{i,s}$$

$$\text{Var}(\hat{\mu}_{i,t}^{\text{esti}}|N_{i,t},\mathcal{W}_{i,t}) = \frac{1}{(N_{i,t})^2} \sum_{s\in\mathcal{W}_{i,t}} \sigma_i^2 = \frac{\sigma_i^2}{N_{i,t}}$$

$$\text{Bias}(\hat{\mu}_{i,t}^{\text{esti}}|N_{i,t},\mathcal{W}_{i,t}) = \frac{1}{N_{i,t}} \sum_{s\in\mathcal{W}_{i,t}} (\mu_{i,s} - \mu_{i,t})$$

Let

$$\text{Bias}(\hat{\mu}_{i,t}^{\text{esti}}) = \mathbb{E}\left[ \frac{1}{N_{i,t}} \sum_{s\in\mathcal{W}_{i,t}} (\mu_{i,s} - \mu_{i,t}) \right]$$

$$\text{Bias}_{\text{cond}} = \mathbb{E}[\hat{\mu}_{i,t}^{\text{esti}}|N_{i,t},\mathcal{W}_{i,t}] - \mu_{i,t}$$

$$= \frac{1}{N_{i,t}} \sum_{s\in\mathcal{W}_{i,t}} (\mu_{i,s} - \mu_{i,t})$$

$$\text{Var}_{\text{cond}} = \text{Var}(\hat{\mu}_{i,t}^{\text{esti}}|N_{i,t},\mathcal{W}_{i,t}) = \frac{\sigma_i^2}{N_{i,t}}$$

Because of $\mu_{i,t}$ is a constant value:

$$\mathbb{E}[(\hat{\mu}_{i,t}^{\text{esti}} - \mu_{i,t})^2|N_{i,t},\mathcal{W}_{i,t}] = \text{Var}_{\text{cond}} + (\text{Bias}_{\text{cond}})^2$$

Because of $\mathbb{E}[\text{Bias}_{\text{cond}}] = \text{Bias}(\hat{\mu}_{i,t}^{\text{esti}})$:

$$MSE(\hat{\mu}^{\text{esti}}, \mu_{i,t}) = \mathbb{E}[\mathbb{E}[(\hat{\mu}_{i,t}^{\text{esti}} - \mu_{i,t})^2|N_{i,t},\mathcal{W}_{i,t}]]$$

$$= \mathbb{E}[\text{Var}_{\text{cond}} + \mathbb{E}[(\text{Bias}_{\text{cond}})^2]$$

$$= \mathbb{E}[\frac{\sigma_i^2}{N_{i,t}}] + \mathbb{E}[(\text{Bias}_{\text{cond}})^2]$$

$$= \mathbb{E}[\frac{\sigma_i^2}{N_{i,t}}] + \text{Var}(\text{Bias}(\hat{\mu}^{\text{esti}})) + (\text{Bias}(\hat{\mu}^{\text{esti}}))^2$$

Based on the definition of $\hat{\mu}_{i,t}^{esti}$(Definition 3), environment change(Definition 1) and $W_{i,t}$ is larger than $t - s$:

$$|Bias(\hat{\mu}_{i,t}^{\text{esti}})| \le \mathbb{E}\left[ \frac{1}{N_{i,t}} \sum_{s\in\mathcal{W}_{i,t}} C(\frac{t-s}{t})^d \right]$$

$$\le \frac{C}{N_{i,t} \cdot t^d} \sum_{x=0}^{W_{i,t}-1} x^d$$

$$\approx \frac{C}{N_{i,t} \cdot t^d} \int_0^{W_{i,t}} x^d dx$$

$$= \frac{C \cdot W_{i,t}^{d+1}}{(d+1) \cdot N_{i,t} \cdot t^d}$$

When algorithms guarantee exploration(Assumption 2):

$$\text{MSE}(\hat{\mu}_{i,t}^{\text{esti}}, \mu_{i,t}) \le \frac{\sigma_i^2 K}{C_{ge} \cdot W_{i,t}} + \frac{C^2 K^2 W_{i,t}^{2d}}{C_{ge}^2(d+1)^2 t^{2d}}$$

$\square$

## A.4 MSE OF PREDICTION MODEL

**Lemma 7.** *Based on the predictor effectiveness(Assumation 1), when algorithms guarantee exploration (Assumption 2), we have:*

$$MSE(\hat{\mu}_{i,t}^{pred}) \leq \frac{C_p K}{C_{gx} \cdot W_{i,t}}$$

*Proof.* With Assumation 1 and Assumation 2, we have:

$$\text{MSE}(\hat{\mu}_{i,t}^{\text{pred}}, \mu_{i,t}) = \mathbb{E}[|\hat{\mu}_{i,t}^{pred} - \mu_{i,t}|^2]$$
$$= C_p \cdot (N_{i,t})^{-1})$$
$$\leq \frac{C_2 K}{C_{ge} \cdot W_{i,t}}$$

$\square$

## A.5 ADAPTIVE WINDOW

**Lemma 8.** *Based on the definition of $\hat{\mu}_{i,t}$(Definition 3), $\hat{\mu}^{esti}$(Formula 6 in the main text) and predictor effectiveness(Assumption 1), we have the optimal window at moment t:*

$$W_{i,t}^* = C_w K^{-1/(2d+1)} t^{2d/(2d+1)}$$

*where $C_w^{2d+1} = \frac{C_{ge}(d+1)^2[(1-\lambda_{i,t})\sigma_i^2 + \lambda_{i,t}C_p]}{2d \cdot (1-\lambda_{i,t})C^2}$*

*Proof.* Our aim is that $L(W_{i,t}, \lambda_{i,t})$ takes the minimum value. Using Appendix A.1, Appendix A.2, Appendix A.3, Appendix A.4, we have:

$$L(W_{i,t}, \lambda_{i,t}) = (1 - \lambda_{i,t})^2 \text{MSE}(\hat{\mu}_{i,t}^{\text{esti}}, \mu_{i,t}) + \lambda_{i,t}^2 \text{MSE}(\hat{\mu}_{i,t}^{\text{pred}}, \mu_{i,t}) + 2\lambda_{i,t}(1 - \lambda_{i,t})\mathbb{E}[\epsilon_e \epsilon_p]$$
$$\leq (1 - \lambda_{i,t})^2 \text{MSE}(\hat{\mu}_{i,t}^{\text{esti}}, \mu_{i,t}) + \lambda_{i,t}^2 \text{MSE}(\hat{\mu}_{i,t}^{\text{pred}}, \mu_{i,t})$$
$$+ \lambda_{i,t}(1 - \lambda_{i,t})(\text{MSE}(\hat{\mu}_{i,t}^{\text{esti}}, \mu_{i,t}) + \text{MSE}(\hat{\mu}_{i,t}^{\text{pred}}, \mu_{i,t}))$$
$$= (1 - \lambda_{i,t})\text{MSE}(\hat{\mu}_{i,t}^{\text{esti}}, \mu_{i,t}) + \lambda_{i,t}\text{MSE}(\hat{\mu}_{i,t}^{\text{pred}}, \mu_{i,t})$$
$$\leq \frac{((1 - \lambda_{i,t})\sigma_i^2 + \lambda_{i,t}C_p)K}{C_{ge} \cdot W_{i,t}} + \frac{(1 - \lambda_{i,t})C^2 K^2 W_{i,t}^{2d}}{C_{ge}^2(d + 1)^2 t^{2d}}$$

To address analytical intractability, we construct a surrogate function $L_{sub}(W_{i,t}, \lambda_{i,t})$. Let:

$$L_{sub}(W_{i,t}, \lambda_{i,t}) = \frac{((1 - \lambda_{i,t})\sigma_i^2 + \lambda_{i,t}C_p)K}{C_{ge} \cdot W_{i,t}} + \frac{(1 - \lambda_{i,t}) \cdot C^2 K^2 W_{i,t}^{2d}}{C_{ge}^2(d + 1)^2 t^{2d}}$$

Taking the partial derivative of $L_{sub}(W, \lambda)$ with respect to $W_{i,t}$, let $\frac{\partial L_{sub}}{\partial W_{i,t}} = 0$:

$$\frac{\partial L_{sub}}{\partial W_{i,t}} = -\frac{((1 - \lambda_{i,t})\sigma_i^2 + \lambda_{i,t}C_2)K}{W_{i,t}^2} + \frac{2d(1 - \lambda_{i,t}) \cdot C^2 K^2 W_{i,t}^{2d-1}}{(d + 1)^2 t^{2d}} = 0$$

At this time, $L_{sub}(W, \lambda)$ takes the minimum value, so that we have:

$$W_{i,t}^* = C_w K^{-1/(2d+1)} t^{2d/(2d+1)}$$

where $C_w^{2d+1} = \frac{C_{ge}(d+1)^2[(1-\lambda_{i,t})\sigma_i^2 + \lambda_{i,t}C_p]}{2d \cdot (1-\lambda_{i,t})C^2}$

$\square$

## A.6 ADAPTIVE WEIGHT

**Lemma 9.** *Based on the definition of $\hat{\mu}_{i,t}$(Definition 3), $\hat{\mu}^{esti}$(Eq. 6) and predictor effectiveness(Assumption 1), we can obtain optimal $\lambda_{i,t}$:*

$$\lambda_{i,t}^* \approx \frac{MSE(\hat{\mu}_{i,t}^{esti}, \mu_{i,t}) - \mathbb{E}[(\hat{\mu}_{i,t}^{esti} - \mu_{i,t})(\hat{\mu}_{i,t}^{pred} - \mu_{i,t})]}{MSE(\hat{\mu}_{i,t}^{esti}, \mu_{i,t}) + MSE(\hat{\mu}_{i,t}^{pred}, \mu_{i,t}) - 2\mathbb{E}[(\hat{\mu}_{i,t}^{esti} - \mu_{i,t})(\hat{\mu}_{i,t}^{pred} - \mu_{i,t})]}$$

*Proof.* Let $A(W_{i,t}) = \text{MSE}(\hat{\mu}_{i,t}^{esti}, \mu_{i,t})$ is the mean square error of estimator errors, $B(W_{i,t}) = \text{MSE}(\hat{\mu}_{i,t}^{pred}, \mu_{i,t})$ is the mean square error of predictor errors, $C(W_{i,t}) = \mathbb{E}[(\hat{\mu}_{i,t}^{esti} - \mu_{i,t})(\hat{\mu}_{i,t}^{pred} - \mu_{i,t})]$ is a combination of estimator error and predictor error. Based on the definition of $\hat{\mu}_{i,t}$(Definition 3), using Appendix A.1, loss function is:

$$\mathcal{L}(W_{i,t}, \lambda_{i,t}) = MSE(\hat{\mu}_{i,t}, \mu_{i,t})$$
$$= (1 - \lambda_{i,t})^2 A(W_{i,t}) + \lambda_{i,t}^2 B(W_{i,t}) + 2\lambda_{i,t}(1 - \lambda_{i,t})C(W_{i,t})$$

Loss function consists of three items: estiory error, predict error and combined error. Taking the partial derivative of $L(W_{i,t}, \lambda_{i,t})$ with respect to $\lambda_{i,t}$, let $\frac{\partial L}{\partial \lambda_{i,t}} = 0$, we have:

$$\lambda_{i,t}^* = \frac{A(W_{i,t}) - C(W_{i,t})}{A(W_{i,t}) + B(W_{i,t}) - 2C(W_{i,t})}$$

$\square$

## A.7 CONFIDENCE BOUND

**Lemma 10.** *Based on the definition of $\hat{\mu}_{i,t}$(Definition 3), $\hat{\mu}^{esti}$(Eq. 6) and predictor effectiveness(Assumption 1), with a probability of at least $1 - \frac{\delta}{KT}$:*

$$|\hat{\mu}_{i,t} - \mu_{i,t}| \le CB_{i,t}(\forall i \in [K], t \in [T])$$

*The confidence bound is:*

$$CB_{i,t} = \sqrt{\frac{2\sigma_i^2 \log(4KT/\delta)}{N_{i,t}}} + Bias_{total}$$

*where*

$$Bias_{total} = ((1 - \lambda_{i,t})C_{e1} + \lambda_{i,t}C_{e2})K^{\frac{d+1}{2d+1}}t^{-\frac{d}{2d+1}}$$
$$C_{e1}^2 = \frac{\sigma_i^2}{C_{ge}C_w} + \frac{C^2 C_w^{2d}}{(d+1)^2 C_{ge}^2}$$
$$C_{e2}^2 = \frac{C_p}{C_{ge}C_w}$$

*Proof.* Step 1: decompose $\hat{\mu}_{i,t} - \mu_{i,t}$

Let:

$$\epsilon_{e,i,t}' = \hat{\mu}_{i,t}^{esti} - \mathbb{E}[\hat{\mu}_{i,t}^{esti}]$$
$$\epsilon_{p,i,t}' = \hat{\mu}_{i,t}^{pred} - \mathbb{E}[\hat{\mu}_{i,t}^{pred}]$$
$$\text{Bias} = (1 - \lambda_{i,t})(\mathbb{E}[\hat{\mu}_{i,t}^{esti}] - \mu_{i,t}) + \lambda(\mathbb{E}[\hat{\mu}_{i,t}^{pred}] - \mu_{i,t})$$

Then:

$$\hat{\mu}_{i,t} - \mu_{i,t} = (1 - \lambda_{i,t})(\hat{\mu}_{i,t}^{esti} - \mu_{i,t}) + \lambda_{i,t}(\hat{\mu}_{i,t}^{pred} - \mu_{i,t})$$
$$= (1 - \lambda_{i,t})(\hat{\mu}_{i,t}^{esti} - \mathbb{E}[\hat{\mu}_{i,t}^{esti}]) + \lambda_{i,t}(\hat{\mu}_{i,t}^{esti} - \mathbb{E}[\hat{\mu}_{i,t}^{esti}])$$
$$+ (1 - \lambda_{i,t})(\mathbb{E}[\hat{\mu}_{i,t}^{esti}] - \mu_{i,t}) + \lambda(\mathbb{E}[\hat{\mu}_{i,t}^{pred}] - \mu_{i,t})$$
$$= (1 - \lambda_{i,t})\epsilon_{e,i,t}' + \lambda_{i,t}\epsilon_{p,i,t}' + \text{Bias}$$

Step 2: calculate$[(1 - \lambda_{i,t})\epsilon'_{e,i,t} + \lambda_{i,t}\epsilon'_{p,i,t}]$

The variance of $[(1 - \lambda_{i,t})\epsilon'_{e,i,t} + \lambda_{i,t}\epsilon'_{p,i,t}]$ is:

$$\text{Var}[(1 - \lambda_{i,t})\epsilon'_{e,i,t} + \lambda_{i,t}\epsilon'_{p,i,t}]$$
$$= (1 - \lambda_{i,t})^2 Var(\epsilon'_{e,i,t}) + \lambda_{i,t}^2 Var(\epsilon'_{p,i,t}) + 2\lambda_{i,t}(1 - \lambda_{i,t})Cov(\epsilon'_{e,i,t}, \epsilon'_{p,i,t})$$

Based on Assumpation 1 and using Hoeffding inequality:

$$\mathbb{P}\left(|\epsilon'_{e,i,t}| > \sqrt{\frac{2\sigma_i^2 \log(4KT/\delta)}{N_{i,t}}}\right) \leq \frac{\delta}{2KT}$$

$$\mathbb{P}\left(|\epsilon'_{p,i,t}| > \sqrt{\frac{2\sigma_i^2 \log(4KT/\delta)}{N_{i,t}}}\right) \leq \frac{\delta}{2KT}$$

So with a probability of at least $1 - \frac{\delta}{2KT}$:

$$Var(\epsilon'_{e,i,t}) \leq \frac{2\sigma_i^2 \log(4KT/\delta)}{N_{i,t}}$$

$$Var(\epsilon'_{p,i,t}) \leq \frac{2\sigma_i^2 \log(4KT/\delta)}{N_{i,t}}$$

Let:

$$\sigma_{e,i,t}^2 = \frac{2\sigma_i^2 \log(4KT/\delta)}{N_{i,t}}$$

$$\sigma_{p,i,t}^2 = \frac{2\sigma_i^2 \log(4KT/\delta)}{N_{i,t}}$$

Based on Cauchy inequality:

$$|Cov(\epsilon'_{e,i,t}, \epsilon'_{p,i,t})| \leq \sigma_{e,i,t}\sigma_{p,i,t}$$

So using Hoeffding inequality again, with a probability of at least $1 - \frac{\delta}{KT}$:

$$\left|(1 - \lambda_{i,t})\epsilon'_{e,i,t} + \lambda_{i,t}\epsilon'_{p,i,t}\right|$$
$$\leq \sqrt{\text{Var}[(1 - \lambda_{i,t})\epsilon'_{e,i,t} + \lambda_{i,t}\epsilon'_{p,i,t}]}$$
$$= \sqrt{(1 - \lambda_{i,t})^2 Var(\epsilon'_{e,i,t}) + \lambda_{i,t}^2 Var(\epsilon'_{p,i,t}) + 2\lambda_{i,t}(1 - \lambda_{i,t})Cov(\epsilon'_{e,i,t}, \epsilon'_{p,i,t})}$$
$$\leq \sqrt{(1 - \lambda_{i,t})^2 \sigma_{e,i,t}^2 + \lambda_{i,t}^2 \sigma_{p,i,t}^2 + 2\lambda_{i,t}(1 - \lambda_{i,t})\sigma_{e,i,t}\sigma_{p,i,t}}$$
$$= \sqrt{\frac{2\sigma_i^2 \log(4KT/\delta)}{N_{i,t}}}$$

Step 3: calculate Bias

The Bias are due to environmental changes and time differences, when algorithms aguarantee exploratory(Assumption 2), using Appendix A.3, Assumation 1:

$$|\mathbb{E}[\hat{\mu}_{i,t}^{\text{esti}}] - \mu_{i,t}| = C_{e1}^2 K^{\frac{2d+2}{2d+1}} t^{-\frac{2d}{2d+1}}$$

$$|\mathbb{E}[\hat{\mu}_{i,t}^{\text{pred}}] - \mu_{i,t}| = C_{e2}^2 K^{\frac{2d+2}{2d+1}} t^{-\frac{2d}{2d+1}}$$

where $C_{e1}^2 = \frac{\sigma_i^2}{C_{ge}C_w} + \frac{C^2 C_w^{2d}}{(d+1)^2 C_{ge}^2}$ and $C_{e2}^2 = \frac{C_p}{C_{ge}C_w}$

So

$$\text{Bias} = (1 - \lambda_{i,t})|\mathbb{E}[\hat{\mu}_{i,t}^{\text{esti}}] - \mu_{i,t}| + \lambda_{i,t}|\mathbb{E}[\hat{\mu}_{i,t}^{\text{pred}}] - \mu_{i,t}|$$

$$= (1 - \lambda_{i,t})C_{e1}K^{\frac{d+1}{2d+1}}t^{-\frac{d}{2d+1}} + \lambda_{i,t}C_{e2}K^{\frac{d+1}{2d+1}}t^{-\frac{d}{2d+1}}$$

$$= ((1 - \lambda_{i,t})C_{e1} + \lambda_{i,t}C_{e2})K^{\frac{d+1}{2d+1}}t^{-\frac{d}{2d+1}}$$

$$= \text{Bias}_{\text{total}}$$

Conclude from the above steps, end of proof. $\qquad\qquad\square$

## A.8 EXPECTIVE CUMULATICE REGRET

**Theorem 2.** *In the non-stationary MAB problem defined in Section 3, the algorithm is used and the optimal parameters $W_{i,t}^*$ selected as in Appenix A.5 and $\lambda_{i,t}^*$ selected as in Appenix A.6. Then the expected cumulative regret satisfies:*

$$\mathbb{E}[R(T)] = O(K^{(3d+2/2d+1)}T^{(d+1)/(2d+1)}(log(KT))^{1/2})$$

*Proof.* step1: Calculate and break down single-step expectation regret: At time $t$, let $i_t^* = \arg\max_i \mu_{i,t}$ be the optimal arm with maximum regard and $I_t$ be the arm selected by the algorithm. The single-step expected regret can be defined as:

$$\mathbb{E}[r_t] = \mathbb{E}[\mu_{i_t^*,t} - \mu_{I_t,t}]$$

$$\leq \sum_{i=1}^{K} \mathbb{P}(I_t = i) \cdot (\mu_{i_t^*,t} - \mu_{i,t})$$

$$\leq \sum_{i=1}^{K} \mathbb{P}(I_t = i) \cdot \Delta_{i,t}$$

Where $\Delta_{i,t} = |\mu_{i_t^*,t} - \mu_{i,t}|$ is the instantaneous gap at time $t$. Let event $D_i$ be confidence interval failure on the arm i and event $E_i$ be confidence intervals are not invalidated on the arm i. We can divide $\sum_{i=1}^{K} P(I_t = i)$ into two cases:

$$\sum_{i=1}^{K} \mathbb{P}(I_t = i) = \sum_{i=1}^{K} \mathbb{P}(\text{D}_i) + \sum_{i=1}^{K} \mathbb{P}(\text{E}_i)$$

When confidence interval failure

$$\sum_{i=1}^{K} \mathbb{P}(\text{D}_i) \leq \sum_{i=1}^{K} \mathbb{P}(|\hat{\mu}_{i,t} - \mu_{i,t}| > \text{CB}_{i,t})$$

$$\leq \sum_{i=1}^{K}(1 - (1 - \frac{\delta}{KT})) = \frac{\delta}{T}$$

When confidence intervals are not invalidated: There are three types of $i$:

(1)$SCORE_{i,t} = SCORE_{i_t^*,t}$ ($i = i_t^*$). In this case, $\mu_{i_t^*,t} - \mu_{i,t} = 0$

(2)$SCORE_{i,t} > SCORE_{i_t^*,t}$. Also $i_t^*$is the optimal arm which means $\mu_{i_t^*,t} > \mu_{i,t}$. So we have:

$$\hat{\mu}_{i,t} - \text{CB}_{i,t} \leq \hat{\mu}_{i^*,t} + \text{CB}_{i^*,t}$$

Therefore, $\Delta_{i,t}$ can be constrained to:

$$\Delta_{i,t} = |\mu_{i_t^*,t} - \mu_{i,t}|$$

$$= |(\mu_{i_t^*,t} - \mu_{i^*,t}) + (\mu_{i^*,t} - \mu_{i_t,t}) + (\mu_{i_t,t} - \mu_{i,t})|$$

$$\leq \text{CB}_{i^*,t} + (\text{CB}_{i^*,t} + \text{CB}_{i,t}) + \text{CB}_{i,t}$$

$$= 2\text{CB}_{i^*,t} + 2\text{CB}_{i,t}$$

$$\sum_{i=1}^{K} \mathbb{P}(E_i) = \sum_{i=1}^{K} \mathbb{P}(\Delta_{i,t} \leq 2CB_{i^*,t} + 2CB_{i,t})$$

(3)$SCORE_{i,t} < SCORE_{i_t^*,t}$. In this case, $\mathbb{P}(I_t = i) = 0$.

Combine three cases, we have:

$$\sum_{i=1}^{K} \mathbb{P}(E_i) \cdot (\mu_{i_t^*,t} - \mu_{i,t}) \leq \sum_{i=1}^{K} \mathbb{P}(\Delta_{i,t} \leq 2CB_{i^*,t} + 2CB_{i,t}) \max_{1 \leq i \leq K}(CB_{i,t})$$

Combine event $D_i$ and event $E_i$:

$$\mathbb{E}[r_t] \leq \sum_{i=1}^{K} [\mathbb{P}(\Delta_{i,t} < 2CB_{i^*,t} + 2CB_{i,t}) \max_{1 \leq i \leq K}(CB_{i,t})] + \frac{\delta}{T}$$

Step 3: Based on Appenix A.7 and Assumption 2, have:

$$O(\max_{1 \leq i \leq K}(CB_{i,t})) = \sqrt{\frac{2\sigma_{max}^2 \log(4KT/\delta)}{N_{i,t}}} + O\left(K^{\frac{d+1}{2d+1}} t^{-\frac{d}{2d+1}}\right)$$

$$= O\left(\sqrt{log(KT)} K^{\frac{d+1}{2d+1}} t^{-\frac{d}{2d+1}}\right)$$

$$\mathbb{E}[r_t] \leq K \cdot O\left(\sqrt{log(KT)} K^{\frac{d+1}{2d+1}} t^{-\frac{d}{2d+1}}\right) + \frac{\delta}{T}$$

$$= O\left(\sqrt{log(KT)} K^{\frac{3d+2}{2d+1}} t^{-\frac{d}{2d+1}}\right) + \frac{\delta}{T}$$

Step 4: Sum all arms:

$$\mathbb{E}[R(T)] = \sum_{t=1}^{T} \mathbb{E}[r_t]$$

$$= \sum_{t=1}^{T} O\left(\sqrt{log(KT)} K^{\frac{3d+2}{2d+1}} t^{-\frac{d}{2d+1}}\right) + \sum_{t=1}^{T} \frac{\delta}{T})$$

$$= O\left(\sqrt{log(KT)} K^{(3d+2)/(2d+1)} \int_1^T t^{-d/(2d+1)} dt\right) + \delta$$

$$\leq O\left(\sqrt{log(KT)} K^{(3d+2)/(2d+1)} \cdot \frac{2d+1}{d+1} T^{(d+1)/(2d+1)}\right) + \delta$$

$$= O\left(K^{(3d+2/2d+1)} T^{(d+1/2d+1)} (log(KT))^{1/2}\right)$$

$\square$

## A.9 THEORETICAL LOWER BOUND

**Lemma 11.** *Based on the environmental change definition 1, the theoretical lower bound of regret is:*

$$\Omega(K^{1/3} T^{1-d/3})$$

*Proof.* Based on the proofed theoretical lower bound $\Omega(K^{1/3} V_t^{2/3} T^{1/3})$Komiyama et al. (2024), according to the determination of $V_t = \sum_{t=1}^{T-1} \sup_i(|\mu_{i,t} - \mu_{i,t-1}|)$, we have:

$$V_t = \sum_{t=1}^{T-1} \sup_i(|\mu_{i,t} - \mu_{i,t-1}|) = O(T^{1-d})$$

So,

$$\Omega(K^{1/3}V_t^{2/3}T^{1/3}) = \Omega(K^{1/3}T^{1-d/3})$$

$\square$

# B PREDICTOR IMPLEMENTATION DETAILS

## B.1 DIFFUSION-BASED CONDITIONAL ARCHITECTURE

The predictor component $f_p(\mathcal{H}_{i,t-1})$ implements a conditional denoising diffusion probabilistic model (DDPM) specifically designed for multi-armed bandit reward prediction. This architecture directly supports Assumption 1 by providing both point estimates $\hat{\mu}_{i,t}^{\text{pred}}$ and uncertainty quantification $\hat{\sigma}_{i,t}^{\text{pred}}$ essential for the adaptive weight mechanism in Lemma 2.

### B.1.1 FORWARD DIFFUSION PROCESS

The forward process corrupts reward observations through a Markov chain with learned variance schedule:

$$q(y_t|y_0) = \mathcal{N}(y_t; \sqrt{\bar{\alpha}_t}y_0, (1 - \bar{\alpha}_t)\mathbf{I})$$

where $\beta_t$ follows a cosine schedule designed for stable training:

$$\beta_t = 1 - \frac{\alpha_t}{\alpha_{t-1}}, \quad \alpha_t = \cos^2\left(\frac{t/T + s}{1 + s} \cdot \frac{\pi}{2}\right)$$

with $s = 0.008$ and $\bar{\alpha}_t = \prod_{i=1}^{t} \alpha_i$. This schedule ensures smooth noise injection while maintaining training stability across the $T = 20$ timesteps.

### B.1.2 CONDITIONAL DENOISING NETWORK

The core denoising network $\epsilon_\theta(y_t, t, \mathcal{H}_{i,t-1}, i)$ incorporates three conditioning mechanisms to capture both temporal dynamics and arm-specific patterns:

**Temporal and Arm Embedding** Time step $t$ and arm index $i$ are jointly embedded through:

$$\text{emb}(t, i) = \text{LayerNorm}(\text{StepEmbed}(t) + \text{ArmEmbed}(i))$$

where both embeddings project to $\mathbb{R}^{128}$, enabling the model to learn arm-specific temporal patterns.

**Historical Context Processing** Reward history $\mathcal{H}_{i,t-1}$ undergoes statistical feature extraction to capture distributional properties:

$$\text{context} = [\mu_{\text{hist}}, \sigma_{\text{hist}}^2, \max(\mathcal{H}_{i,t-1}), \min(\mathcal{H}_{i,t-1}), Q_{25}, Q_{75}]$$

These features provide robust statistical summaries that remain informative even with limited observations, directly supporting the $C_{ge} \cdot N_{i,t}^{-1}$ term in Assumption 1.

**Dual-Output Architecture** The network produces both noise predictions and uncertainty estimates:

$$\epsilon_{\text{pred}}, \sigma_{\text{pred}} = \text{DenoiseNet}([\text{emb}(t, i), \text{context}, y_t])$$

$$\hat{\mu}_{i,t}^{\text{pred}} = \frac{1}{\sqrt{\alpha_t}}\left(y_t - \frac{\beta_t}{\sqrt{1 - \bar{\alpha}_t}}\epsilon_{\text{pred}}\right)$$

$$\hat{\sigma}_{i,t}^{\text{pred}} = \text{Softplus}(\sigma_{\text{pred}})$$

## B.2 ARM-SPECIFIC MODELING STRATEGY

To address the fundamental challenge in multi-armed bandits where observations are sparse and unevenly distributed across arms ($N_{i,t} \ll t$ and $N_{i,t} \neq N_{j,t}$), the implementation employs arm-independent modeling.

### B.2.1 INDEPENDENT HISTORY MANAGEMENT

Each arm $i \in \{1, \ldots, K\}$ maintains separate data structures:

$$\text{hist\_buffers}[i] \in \mathbb{R}^{\text{hist\_len}} \quad \text{(circular buffer)}$$
$$\text{step\_counters}[i] \in \mathbb{N} \quad \text{(arm-specific time)}$$
$$\text{training\_buffers}[i] \subset \mathbb{R}^{\text{window\_size}} \times \mathbb{R} \quad \text{(experience replay)}$$

This design ensures that prediction quality for arm $i$ depends primarily on $N_{i,t}$ rather than global time $t$, directly implementing the theoretical dependency structure in Assumption 1:

$$\mathbb{E}[|\hat{\mu}_{i,t}^{\text{pred}} - \mu_{i,t}|^2] \leq C_{ge} \cdot N_{i,t}^{-1}$$

### B.2.2 ADAPTIVE WINDOW SELECTION

The predictor automatically adjusts window size based on available arm-specific data:

$$\text{window\_size}[i] = \begin{cases} \min(W_{\text{default}}, |\text{hist\_buffers}[i]|) & \text{if } N_{i,t} \geq 5 \\ |\text{hist\_buffers}[i]| & \text{otherwise} \end{cases}$$

where $W_{\text{default}} = 20$. This mechanism prevents overfitting when $N_{i,t}$ is small while maximizing information utilization when sufficient observations exist.

## B.3 ONLINE LEARNING AND VARIANCE CALIBRATION

### B.3.1 INCREMENTAL TRAINING PROTOCOL

The predictor supports continuous adaptation through mini-batch updates on recent experiences. For each arm $i$, training samples are collected as:

$$\mathcal{S}_i = \{(\mathcal{H}_{i,s}, r_{i,s+1}) : s \in \mathcal{T}_i, |\mathcal{H}_{i,s}| \geq 5\}$$

where $\mathcal{T}_i$ represents time steps when arm $i$ was selected.

The training objective minimizes the denoising loss:

$$\mathcal{L}_i = \mathbb{E}_{(\mathcal{H},r) \in \mathcal{S}_i, t \sim \mathcal{U}(0,T), \epsilon \sim \mathcal{N}(0,\mathbf{I})}[\|\epsilon - \epsilon_\theta(y_t, t, \mathcal{H}, i)\|^2]$$

where $y_t = \sqrt{\bar{\alpha}_t} r + \sqrt{1 - \bar{\alpha}_t} \epsilon$ represents the noised target reward.

### B.3.2 VARIANCE CORRECTION MECHANISM

Neural diffusion models often exhibit systematic bias in uncertainty estimation. To address this, we implement adaptive variance correction:

$$\text{correction\_factor}[i]_t = \gamma \cdot \text{correction\_factor}[i]_{t-1} + (1 - \gamma) \cdot \frac{\text{Var}(\mathcal{H}_{i,t})}{\hat{\sigma}_{\text{pred}}^2}$$

where $\gamma = 0.95$ is the exponential moving average decay factor. The corrected variance estimate becomes:

$$\hat{\sigma}_{i,t}^{\text{pred, corrected}} = \hat{\sigma}_{i,t}^{\text{pred}} \cdot \sqrt{\text{correction\_factor}[i]_t}$$

This correction is crucial for accurate $\lambda_{i,t}$ computation in Lemma 2, as the optimal mixing weight depends on the relative mean squared errors of estimation and prediction components.

## B.4 THEORETICAL JUSTIFICATION AND COMPUTATIONAL ANALYSIS

### B.4.1 SUPPORTING ASSUMPTION 1

The implementation architecture directly realizes the theoretical assumption through two mechanisms:

**Data-Driven Improvement** Arm-specific modeling ensures that prediction quality improves with increasing $N_{i,t}$, implementing the $C_{ge} \cdot N_{i,t}^{-1}$ term. Each additional observation of arm $i$ directly contributes to better predictions for that arm, independent of other arms' observation frequencies.

**Uncertainty Quantification** The dual-output architecture provides principled uncertainty estimates essential for computing optimal mixing weights $\lambda_{i,t}^*$ in Lemma 2. The variance correction mechanism ensures these estimates accurately reflect true prediction confidence.

### B.4.2 COMPUTATIONAL COMPLEXITY ANALYSIS

The predictor maintains favorable computational properties:

- **Space Complexity**: $O(K \cdot \text{hist\_len})$ for independent arm buffers
- **Time Complexity per Prediction**: $O(1)$ with respect to $K$ and $t$
- **Training Complexity**: $O(\text{batch\_size} \cdot \text{network\_depth})$ per update

This ensures scalability with the number of arms while providing the distributional forecasts essential for exploration-exploitation balance in the UEP framework.

### B.5 EXPERIMENTAL ENVIRONMENTS

This section provides detailed mathematical definitions of the experimental environments used to evaluate UEP across different non-stationarity scenarios. All environments operate with $K = 4$ arms over $T = 1000$ time steps, with rewards constrained to the range $[0.1, 0.9]$ to ensure meaningful exploration-exploitation trade-offs.

### B.5.1 STATIC ENVIRONMENT

The static environment serves as a baseline with fixed reward distributions and minimal noise:

$$\mu_{i,t} = \text{clip}(\mu_i^{\text{base}} + \eta_{i,t}, 0.1, 0.9)$$

where $\mu_i^{\text{base}} = [0.35, 0.75, 0.45, 0.60]$ for arms $i \in \{1, 2, 3, 4\}$ respectively, and $\eta_{i,t} \sim \mathcal{N}(0, 0.02^2)$ represents measurement noise. This environment has $d = \infty$ (stationary) and provides a clear optimal arm (arm 2) to test algorithm performance in the absence of environmental changes.

### B.5.2 SMOOTH NON-STATIONARY ENVIRONMENT ($d = 1.2$)

The smooth environment features learnable periodic patterns with gradual temporal decay:

$$\mu_{i,t} = \text{clip}(\mu_i^{\text{base}} + P_{i,t} + G_{i,t} + N_{i,t}, 0.1, 0.9)$$

$$P_{i,t} = 0.25 \left[ 0.6 \sin\left(\frac{2\pi t}{120 + 15i}\right) + 0.3 \sin\left(\frac{2\pi t}{200 + 25i}\right) + 0.1 \cos\left(\frac{2\pi t}{80 + 10i}\right) \right]$$

$$G_{i,t} = 0.15 \sin\left(\frac{2\pi t}{300} + \frac{i\pi}{2}\right)$$

$$N_{i,t} = 0.1 \cdot \mathcal{N}(0, 1) \cdot (t + 1)^{-d/2}$$

where $\mu_i^{\text{base}} = [0.3, 0.7, 0.4, 0.6]$, $P_{i,t}$ represents multi-frequency periodic patterns, $G_{i,t}$ introduces global trends causing arms to rotate optimal status, and $N_{i,t}$ provides temporally decaying noise. This environment satisfies Definition 1 with $d = 1.2$ and tests the predictor's ability to learn temporal patterns.

### B.5.3 ABRUPT CHANGE ENVIRONMENT ($d = 0.8$)

The abrupt environment introduces discrete regime shifts at predetermined change points:

$$\mu_{i,t} = \text{clip}(\phi_{\text{phase}(t),i} + A_{i,t} + \epsilon_{i,t}, 0.1, 0.9)$$

where $\text{phase}(t) = \lfloor 5t/T \rfloor$ determines the current regime, and the phase-specific base rewards are:

$$\Phi = \begin{bmatrix} 0.7 & 0.3 & 0.5 & 0.4 \\ 0.4 & 0.6 & 0.3 & 0.7 \\ 0.6 & 0.4 & 0.8 & 0.2 \\ 0.3 & 0.7 & 0.4 & 0.6 \\ 0.8 & 0.5 & 0.3 & 0.7 \end{bmatrix}$$

The adaptive noise terms are:

$$A_{i,t} = 0.1 \cdot (t+1)^{-d} \sin\left(\frac{t}{30} + \frac{i\pi}{3}\right)$$

$$\epsilon_{i,t} = 0.08 \cdot \mathcal{N}(0,1)$$

Change points occur at $t \in \{T/5, 2T/5, 3T/5, 4T/5\}$, creating sudden distribution shifts that test change detection capabilities.

### B.5.4 HIGH FREQUENCY ENVIRONMENT ($d = 0.6$)

The high frequency environment exhibits rapid oscillations with temporal decay:

$$\mu_{i,t} = \text{clip}(\mu_i^{\text{base}} + H_{i,t} + R_{i,t}, 0.1, 0.9)$$

$$H_{i,t} = 0.2 \left[0.5 \sin\left(\frac{2\pi t}{25 + 5i}\right) + 0.3 \sin\left(\frac{2\pi t}{45 + 8i}\right) + 0.2 \cos\left(\frac{2\pi t}{70 + 12i}\right)\right]$$

$$R_{i,t} = 0.1 \cdot \mathcal{N}(0,1) \cdot (t+1)^{-d}$$

where $\mu_i^{\text{base}} = [0.4, 0.6, 0.3, 0.7]$. The high-frequency components $H_{i,t}$ create rapid reward oscillations that challenge algorithms requiring fast adaptation, while the decay factor $(t+1)^{-d}$ with $d = 0.6$ ensures the changes remain substantial throughout the horizon.

### B.5.5 COMPETITIVE BALANCED ENVIRONMENT ($d = 0.7$)

The competitive environment ensures arms cyclically achieve optimal status through structured competition:

$$\mu_{i,t} = \text{clip}(\mu_i^{\text{base}} + C_{i,t} + W_t + L_{i,t} + \xi_{i,t}, 0.1, 0.9)$$

$$C_{i,t} = 0.15 \sin\left(\frac{2\pi t}{80 + 15i} + \frac{i\pi}{2}\right)$$

$$W_t = 0.08 \sin\left(\frac{2\pi t}{200}\right)$$

$$L_{i,t} = 0.12 \cdot (t+1)^{-d} \sin\left(\frac{t}{40} + \frac{i\pi}{3}\right)$$

$$\xi_{i,t} = 0.08 \cdot \mathcal{N}(0,1)$$

where $\mu_i^{\text{base}} = [0.45, 0.55, 0.4, 0.6]$, $C_{i,t}$ creates arm-specific advantage cycles, $W_t$ introduces global fluctuations, $L_{i,t}$ provides local perturbations, and $\xi_{i,t}$ adds independent noise. The phase-shifted sinusoidal terms ensure balanced competition where each arm achieves temporary optimality.

### B.5.6 ENVIRONMENT ANALYSIS METRICS

To quantify environment difficulty and validate experimental design, we define several analysis metrics:

**Instantaneous Gap** The gap between the best and second-best arms at time $t$:

$$\Delta_t = \max_i \mu_{i,t} - \max_{i \neq \arg\max_j \mu_{j,t}} \mu_{i,t}$$

**Optimal Arm Switches** The number of times the optimal arm changes:

$$S = \sum_{t=2}^{T} \mathbf{1}[\arg\max_i \mu_{i,t} \neq \arg\max_i \mu_{i,t-1}]$$

**Overlap Ratio**    The normalized overlap between arm reward ranges:

$$O = \frac{1}{T} \sum_{t=1}^{T} \left( 1 - \frac{\max_i \mu_{i,t} - \min_i \mu_{i,t}}{0.8} \right)$$

**Environment Difficulty Score**    A composite measure combining gap size and variability:

$$\mathcal{D} = \frac{1}{\bar{\Delta} + \epsilon} \cdot \left( 1 + \frac{\mathrm{Var}(\Delta_t)}{\bar{\Delta}^2} \right)$$

where $\bar{\Delta} = \frac{1}{T} \sum_{t=1}^{T} \Delta_t$ and $\epsilon = 10^{-6}$ prevents division by zero.

These environments collectively test different aspects of non-stationary bandit algorithms: the smooth environment validates predictive capabilities, the abrupt environment tests change detection, the high-frequency environment challenges rapid adaptation, and the balanced environment evaluates performance under competitive arm dynamics. All environments maintain complexity through their mathematical construction while remaining within the theoretical framework of Definition 1.

### B.6    IMPLEMENTATION ROBUSTNESS

#### B.6.1    GRACEFUL DEGRADATION

For arms with insufficient historical data ($N_{i,t} < 5$), the predictor employs conservative fallback:

$$\hat{\mu}_{i,t}^{\mathrm{pred}} = 0.5, \quad \hat{\sigma}_{i,t}^{\mathrm{pred}} = 0.1 \quad \text{when } N_{i,t} < 5$$

This uniform prior with low confidence ensures that the adaptive weight mechanism in Lemma 2 appropriately reduces reliance on prediction ($\lambda_{i,t} \to 0$) when insufficient data exists.

#### B.6.2    MEMORY MANAGEMENT

Fixed-size circular buffers with hist_len $= 100$ and training_buffer_size $= 500$ prevent unbounded memory growth while retaining recent relevant history. These sizes are calibrated based on typical environmental change rates and ensure sufficient data for pattern detection while maintaining computational efficiency.

#### B.6.3    NUMERICAL STABILITY

The implementation includes several numerical safeguards:

- Gradient clipping during training to prevent exploding gradients
- Minimum variance thresholding in variance correction to avoid division by zero
- Softplus activation for variance prediction to ensure positive outputs
- Cosine noise schedule clipping to $[0, 0.999]$ for training stability

These measures ensure robust performance across diverse environmental conditions while maintaining the theoretical guarantees established in Theorem 1.

### B.7    BASELINES

### B.8    ENVIRONMENT ESTIMATION

Here, we check the correctness of the $d$ estimation, which determines both window sizing and mixing weight computation through Lemma 1 and 2. Figure 3 shows estimation accuracy with a correlation coefficient of 0.974 and MAE of 0.056 across $d \in [0.3, 1.0]$, the fitting effect is relatively good. When $d > 1.0$ approaches stationarity where change detection fails. Inaccurate $d$ estimation propagates through window sizing to MSE computation, directly affecting $\lambda_{i,t}$ selection and algorithm performance.

| Rank | Algorithm | Static | Smooth | Abrupt | High-Freq | Balanced | Avg Rank |
|------|-----------|--------|--------|--------|-----------|----------|----------|
| 1 | **UEP** | 16.9±0.6 | **39.1±2.1** | **35.7±7.4** | 22.8±0.6 | 76.4±7.1 | **1.80** |
| 2 | Variance-Aware-UCB | 22.3±0.2 | 40.6±0.6 | 52.9±4.1 | 27.2±0.8 | 76.9±3.2 | 3.80 |
| 3 | MOSS | 24.2±0.4 | 41.0±0.5 | 57.7±4.4 | 32.5±0.4 | **71.3±3.9** | 4.20 |
| 4 | KL-UCB | 36.3±0.3 | 41.6±0.5 | 59.8±2.6 | 34.1±0.4 | 79.2±2.8 | 6.00 |
| 5 | ADS-KL-UCB | 36.3±0.3 | 41.8±0.5 | 61.1±2.6 | 34.2±0.4 | 78.3±3.5 | 6.60 |
| 6 | Change-Point-Detection | 17.8±0.1 | 62.8±2.4 | 49.1±17.6 | 47.6±1.9 | 90.9±9.6 | 6.60 |
| 7 | Optimistic-Initialization | **8.9±0.0** | 52.2±0.2 | 188.0±8.7 | **15.0±0.5** | 115.0±12.8 | 7.00 |
| 8 | Explore-Then-Commit | 10.7±0.2 | 64.1±0.2 | 198.4±2.0 | 25.2±0.2 | 96.8±25.0 | 8.20 |
| 9 | Sliding-Window-UCB | 33.4±0.3 | 63.2±0.3 | 95.2±9.8 | 42.7±0.4 | 92.4±3.0 | 9.40 |
| 10 | Adaptive-Epsilon-Greedy | 21.5±2.5 | 70.7±2.4 | 199.3±4.6 | 34.5±2.9 | 84.3±10.0 | 9.60 |
| 11 | UCB1 | 65.7±0.4 | 59.4±0.3 | 75.1±3.0 | 57.0±0.3 | 87.8±2.0 | 9.80 |
| 12 | Beta-SWTS | 18.5±1.0 | 81.0±3.2 | 160.8±18.0 | 39.0±1.6 | 97.1±3.8 | 9.80 |
| 13 | CUSUM-UCB | 66.0±1.6 | 84.3±1.0 | 60.7±8.6 | 56.9±0.3 | 100.1±2.0 | 11.60 |
| 14 | Adaptive-Window-UCB | 131.2±5.5 | 87.7±5.9 | 78.2±5.6 | 114.3±9.9 | 111.6±6.9 | 14.60 |
| 15 | Gamma-SWGTS | 54.8±1.6 | 113.6±4.3 | 193.5±9.2 | 67.7±3.8 | 116.9±4.1 | 15.00 |
| 16 | Gradient-Bandit | 56.0±3.4 | 106.7±4.2 | 222.2±4.9 | 68.7±2.9 | 115.3±5.3 | 15.40 |
| 17 | Epsilon-Greedy | 96.8±79.6 | 130.4±64.6 | 193.2±54.5 | 66.4±58.4 | 116.0±49.4 | 15.60 |
| 18 | Discounted-UCB | 130.6±0.5 | 121.8±0.2 | 134.8±1.4 | 117.1±0.3 | 118.2±1.0 | 16.60 |
| 19 | EXP3 | 103.8±15.9 | 147.2±14.6 | 228.5±5.4 | 108.6±9.9 | 133.2±10.1 | 18.20 |
| 20 | Softmax-Bandit | 189.8±3.8 | 227.5±5.1 | 232.3±5.6 | 188.6±5.6 | 161.8±7.7 | 20.00 |
| 21 | Adaptive-Greedy | 213.6±3.6 | 251.5±6.1 | 234.2±5.4 | 214.5±4.7 | 166.9±5.3 | 21.20 |
| 22 | ADS-Thompson-Sampling | 212.6±1.5 | 251.6±4.7 | 251.1±13.1 | 215.1±2.3 | 167.7±2.9 | 22.20 |
| 23 | ADR-Thompson-Sampling | 216.7±24.5 | 257.5±22.4 | 238.3±8.4 | 216.4±16.0 | 167.4±10.0 | 22.60 |

Table 4: Baseline Comparison: Cumulative Regret (Mean±Std)

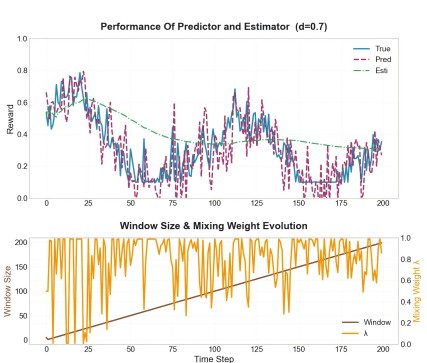

Figure 2: Pred and Esti Performance and adaptive parameters

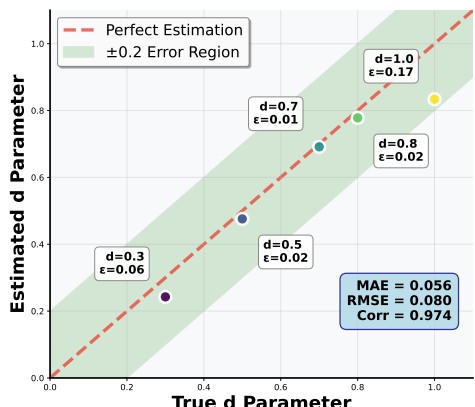

Figure 3: Environmental parameter estimation.

### B.8.1 UNIFIED FRAMEWORK ADAPTIVITY AND PREDICTOR PERFORMANCE

The adaptive weighting mechanism of UEP in Lemma 2 can automatically select between statistical estimation and prediction enhancement paradigms without requiring prior environmental knowledge. To verify this theory, we analyze the behavior of UEP in a smooth single-arm environment ($d = 0.7$) as an example.

Figure 2 shows three key validation results. First, the diffusion-based predictor can accurately capture reward patterns with small errors after the estimation of $d$ is completed, while the statistical estimator, despite tracking the overall trend, exhibits larger deviations. Second, when prediction errors are small, $\lambda_{i,t}$ maintains high values, automatically emphasizing prediction information over statistical estimation, as indicated by Lemma 2. Third, the adaptive window size $W_{i,t}$ grows monotonically in accordance with Lemma 1.

