# OpenReview forum: "UEP: Unifying Estimation and Prediction for Non-stationary Multi-armed Bandits"
_ICLR.cc/2026/Conference — Submitted to ICLR 2026_

### Official Review · Reviewer_FAWN · 2025-10-20

**Soundness:** 2
**Presentation:** 2
**Contribution:** 2
**Rating:** 2
**Confidence:** 3

**Summary:**

Previous studies on non-stationary multi-armed bandits (MAB) typically consider two main settings of non-stationarity and corresponding measures: the drifting case, quantified by the total variation budget, and the piecewise-stationary case, characterized by the number of change points. Both approaches focus on the _cumulative_ non-stationarity of the environment, but ignore _local smoothness_ of the reward dynamics.

This paper instead introduces a smoothness-based assumption, which restricts how sharply the expected rewards can vary locally over time. Under this smoother and stronger assumption, the algorithm can not only mitigate non-stationarity by _forgetting outdated data_ (as in sliding-window methods), but also _leverage predictable temporal trends_ through a prediction module.

By adaptively weighting the estimation and prediction components, the method aims to achieve a more accurate estimate of the current reward mean.

**Strengths:**

- The theoretical analysis appears solid and carefully constructed.
- The idea of combining estimation and prediction in non-stationary bandits is novel and interesting.

**Weaknesses:**

- The non-stationarity assumption might be overly strong or even deliberately designed to justify the use of a predictor, rather than addressing a fundamental and widely relevant bandit setting. As a result, the problem formulation may lack generality and theoretical significance.
- The performance guarantee of the predictor critically relies on Assumption 1, which assumes its MSE decreases as O(1/N). However, the paper does not provide any rigorous proof that the diffusion-based predictor actually satisfies this assumption.
- In the abstract, the authors claim that their bound “improves upon existing results.” Yet, the “existing result” they refer to does not actually exist in the same setting—it is derived by re-expressing prior total-variation-based bounds (e.g., Besbes et al., 2014) in terms of the new smoothness parameter $d$. Therefore, the claimed improvement is not a fair or directly comparable result. It does not convincingly demonstrate that the predictor yields a genuine theoretical benefit, since the benchmark methods were not optimized for this new smoothness-based assumption.

**Questions:**

I am uncertain about the fundamental value of the problem formulation itself.

Is this truly a meaningful extension of the non-stationary bandit problem, or is it a specifically constructed setting to justify introducing a predictor? If it is the latter, the overall contribution becomes less convincing.

Moreover, even after adopting a much stronger smoothness assumption on the environment, the algorithm still needs an additional Assumption 1 to guarantee theoretical validity of the predictor. This weakens the practical value of incorporating the predictor: it appears that the theoretical improvement mainly results from the assumption, rather than from the predictive model itself.

Ideally, the work would have been more impactful if it had shown improvements under the same assumptions as previous non-stationary bandit works, purely due to the introduction of the prediction mechanism.

Under the current formulation, the contribution feels somewhat circular and assumption-driven rather than insight-driven.

---

### Official Review · Reviewer_6ajK · 2025-10-25

**Soundness:** 3
**Presentation:** 3
**Contribution:** 3
**Rating:** 6
**Confidence:** 2

**Summary:**

In this paper, the authors proposed and studied the UEP framework, a framework for non-stationary multi-armed bandits that jointly leverages statistical estimation and predictive modeling to adapt to evolving environments. Theoretically, the authors derived optimal window sizes, adaptive weights, and confidence bounds under their framework, yielding the regret upper bound analysis of their method.
Numerically, they also conducted extensive numerical experiments comparing their method against a number of baselines, demonstrating the efficacy of their method.

**Strengths:**

- Overall, I think this paper is well written and organized, studying a novel problem in the field of non-stationary bandits. The idea of unifying estimation and prediction components through adaptive weighting is conceptually novel.
- The theoretical results appear solid (though I have not checked the proofs in details).
- The numerical experiments are extensive and comprehensive. The paper benchmarks against a broad suite of baselines and also includes ablation studies that show the values of each of the algorithm's components.

**Weaknesses:**

- The diffusion-based predictor can introduce some computational overhead. Could the authors comment on the method’s complexity, especially regarding its scalability to more realistic, higher-dimensional settings? Is the algorithm sensitive to the choice of diffusion model and its hyperparameters?
- While the synthetic experiments are extensive, I think it'd be more helpful to demonstrate the algorithm's performance on some real-world, time-varying data to show the practical relevance of the proposed method. In particular, real-world environments can be adversarial or non-ergodic and some of the key assumptions may fail to hold. I wonder whether the proposed framework/method would remain effective and robust under such settings.
- [Minor] The paper presents the regret analysis before fully introducing the algorithm, which makes the flow somewhat counterintuitive.

**Questions:**

See weaknesses.

---

### Official Review · Reviewer_DxPm · 2025-10-26

**Soundness:** 2
**Presentation:** 2
**Contribution:** 2
**Rating:** 2
**Confidence:** 4

**Summary:**

The authors provides a new algorithm for MAB setting with non-stationary rewards. They assume that the behaviour of the nonstationarity has some regularity properties, novel in the literature, and derived an algorithm UEP that combines reward estimation over an adaptive time window with an estimation module to obtain regret bounds. They tested their algorithm over synthetically generated settings.

**Strengths:**

- clear structure of the paper
- novel approach for a standard setting

**Weaknesses:**

- The writing should be polished, and the formalism of the paper should be improved
- The algorithm relies on parameters that are not known to the learner in standard settings
- Assumptions are not explained properly
- The experiments lack some comparison with the SOTA

**Questions:**

1) You should provide some examples in which the two assumptions hold or characterize the set of predictors that satisfy this assumption. Otherwise, it would not be possible to understand the limitations of your method.
2) The optimal parameter lambda^* is unknown to the learner, but it seems that to achieve the regret you provided, it is necessary to run your algorithm using this value. How would it be possible?
3) The definition of CB highlights that the parameter d should be known a priori to the learner. I think this is a strong assumption that should be stated and discussed in the problem formulation.
4) A more detailed discussion on how the result in Theorem 1 is related to other stationary and non-stationary regret bounds results is necessary.
5) You should define N_it more formally
7) I would have appreciated a more detailed description of the high-level algorithm
8) There also exists a version of SWTS not using an adaptive SW that is worth comparing with your method.
9) How did you set d in your algorithm, and how did you compute d in the environments?

---

### Official Review · Reviewer_Rckj · 2025-11-10

**Soundness:** 2
**Presentation:** 3
**Contribution:** 2
**Rating:** 2
**Confidence:** 3

**Summary:**

The paper addresses non-stationary multi-armed bandit (MAB) problems, where reward distributions change over time. The proposed UEP framework unifies a backward-looking statistical estimator, based on adaptive sliding windows, with a forward-looking predictor implemented via a diffusion-based probabilistic time series model. An adaptive weight $\lambda_{i,t}$ dynamically balances the two components according to their estimated mean squared errors, without needing prior knowledge of the environment's change rate $d$.

The framework derives optimal window sizes and mixing weights theoretically, leading to regret bounds of $O(K^{(3d+2)/(2d+1)} T^{(d+1)/(2d+1)} (\log(KT))^{1/2})$, which improve upon existing bounds like $O(K^{1/3} T^{1-d/3})$ for fast-changing environments ($d < 1$). Evaluations across synthetic environments demonstrate performance advantages over baselines, with ablations validating the components.

**Strengths:**

- **Novel Integration**: The fusion of diffusion-based predictive modeling into non-stationary MABs represents a creative combination of statistical estimation and time series forecasting.

- **Theoretical Rigor**: The paper provides detailed derivations for optimal parameters and regret bounds, including proofs in appendices, that outperform prior methods in rapid-change scenarios.

- **Clear Structure**: The presentation is logical, with well-defined problem formulation, pseudocode, and discussions of modules like the window-based estimator and diffusion predictor.

**Weaknesses:**

- **Gaps in Literature Coverage**: The paper overlooks related work on time series-integrated bandits, such as periodic non-stationary models (e.g., "Non-Stationary Bandits with Periodic Variation" at IFAAMAS 2024 or "Harnessing Ramanujan Periodicity Transforms to Conquer Time-Varying Bandits" in ICASSP 2024), which could better contextualize the motivation for learnable patterns and impact novelty claims.

- **Assumptions validation with real examples**: Assumptions 1 and 2 (e.g., predictor error bounded by $1/N_{i,t}$ and guaranteed exploration) are theoretically standard but potentially unrealistic in data-sparse regimes; A direct derivation of how these assumptions will remain valid despite data-hungry predictors are in place would be a necessary add.

- **Insufficient Justification for Design Choices**: The selection of diffusion models is motivated for forecasting but lacks direct comparison against alternatives like transformers or AR models. The decision to go with diffusion models is not completely justified.

- **Counterintuitive Use of Data-Hungry Models**: Bandit algorithms emphasize data efficiency for rapid exploration-exploitation, yet the choice of diffusion-based predictors, which typically require significant training data, appears counterintuitive. This relates to Assumptions 1 and 2, which rely on ample per-arm observations; without real-world validation in sparse-data scenarios, the framework's practicality remains unclear.

- **Limited Exploitation of Real-World Structures**: A core motivation for real-world non-stationary MAB modeling is the potential to exploit additional structures like temporal dependencies or causality, yet UEP focuses on general prediction without comparisons to specialized methods (e.g., AR bandits or structural causal bandits). UEP being superior or at par in performance under a structured scenario against a specialized methodology is a curious case study that would be a good addition.

- **Unhandled Constants and Scope Restrictions**: Numerous theoretical constants (e.g., $C_p$, $C_{ge}$) lack analysis of parameter impacts or controls.

**Questions:**

Please look at the Weakness section for this

---

### Meta-Review · Area_Chair_nsjN · 2026-01-04

**Summary:**

This submission proposes UEP, a unified framework for non-stationary multi-armed bandits that combines (i) a backward-looking statistical estimator based on adaptive sliding windows and (ii) a forward-looking predictor (implemented as a diffusion-based probabilistic time-series model). The method uses an adaptive weight to balance estimator vs. predictor based on estimated errors, claims improved regret bounds in fast-changing regimes (under additional assumptions), and supports the proposal with synthetic experiments.

Across reviews, the main concerns are that the paper’s problem formulation and assumptions may be overly strong / not well-justified, the diffusion predictor’s key guarantee is assumed rather than proved, and the practical relevance is unclear due to unknown/strong parameters, computational overhead, and lack of real-world validation / missing key comparisons.

**Reviewer Concerns:**

Since the authors did not provide a rebuttal, none of the concerns were addressed.

**Reviewer Scores:**

Since the authors did not provide a rebuttal, the reviewers’ scores would not have increased.

---

### Decision · Program_Chairs · 2026-01-26

Reject